# POMONAG: Pareto-Optimal Many-Objective Neural Architecture Generator

## Abstract

Neural Architecture Search (NAS) automates the design of neural network architectures, minimising dependence on human expertise and iterative experimentation. While NAS methods are often computationally intensive and dataset-specific, employing auxiliary predictors to estimate architecture properties has proven extremely beneficial. These predictors substantially reduce the number of models requiring training, thereby decreasing overall search time. This strategy is frequently utilised to generate architectures satisfying multiple computational constraints. Recently, Transferable Neural Architecture Search (Transferable NAS) has emerged, generalising the search process from being dataset-dependent to task-dependent. In this domain, DiffusionNAG stands as a state-of-the-art method. This diffusion-based method streamlines computation, generating architectures optimised for accuracy on unseen datasets without the need for further adaptation. However, by concentrating exclusively on accuracy, DiffusionNAG neglects other crucial objectives like model complexity, computational efficiency, and inference latency – factors essential for deploying models in resource-constrained, real-world environments. This paper introduces the Pareto-Optimal Many-Objective Neural Architecture Generator (POMONAG), extending DiffusionNAG through a many-objective diffusion process. POMONAG simultaneously considers accuracy, the number of parameters, multiply-accumulate operations (MACs), and inference latency. It integrates Performance Predictor models to estimate these secondary metrics and guide the diffusion gradients. POMONAG's optimisation is enhanced by expanding its training Meta-Dataset, applying Pareto Front Filtering to generated architectures, and refining embeddings for conditional generation. These enhancements enable POMONAG to generate Pareto-optimal architectures that outperform the previous state-of-the-art in both performance and efficiency. Results were validated on two distinct search spaces – NASBench201 and MobileNetV3 – and evaluated across 15 image classification datasets.

## 1 Introduction

Deep learning has become indispensable in various domains by enabling models to learn intricate patterns from large datasets. The architecture of neural networks plays a pivotal role in their performance, traditionally requiring expert knowledge and extensive experimentation. Neural Architecture Search (NAS) automates this design process, aiming to discover optimal architectures without human intervention. However, conventional NAS methods often involve significant computational costs and are typically tailored to specific datasets, limiting their scalability and general applicability. Transferable Neural Architecture Search (Transferable NAS) addresses these limitations by generalising the search process across different tasks. DiffusionNAG stands out in this domain, utilising diffusion processes to generate neural architectures optimised for accuracy on unseen datasets. While effective in reducing computational overhead, DiffusionNAG focuses solely on maximising accuracy, neglecting other crucial performance metrics such as model size, computational cost, and inference latency. In practical applications, especially within resource-constrained environments like mobile devices and embedded systems, it is essential to consider multiple objectives simultaneously. Existing Multi- and Many-objective NAS methods often face scalability challenges or require extensive computational resources, making them less practical for widespread adoption.

This work introduces the Pareto-Optimal Many-Objective Neural Architecture Generator (POMONAG), which extends the capabilities of DiffusionNAG by performing a many-objective optimisation. POMONAG simultaneously considers accuracy, the number of parameters, multiply-accumulate operations (MACs), and inference latency during the architecture generation process. Auxiliary Performance Predictor models, trained to estimate these metrics, are integrated into the diffusion process, guiding exploration towards regions of the search space offering optimal trade-offs among these objectives. This approach facilitates the generation of architectures that are both highly accurate and efficient in computational resources and speed. To enhance POMONAG's performance and adaptability, several key improvements have been implemented. The training Meta-Dataset has been significantly expanded, incorporating a diverse set of architectures and tasks to enhance the Performance Predictors' capabilities across different domains. Pareto Front Filtering and Stretching techniques are applied to balance the multiple objectives effectively, ensuring that the generated architectures are Pareto-optimal. Additionally, the embeddings used for conditional generation have been refined to facilitate more accurate and dataset-aware architecture synthesis. Extensive experiments validate POMONAG's effectiveness. Evaluations conducted on two prominent search spaces—NASBench201 and MobileNetV3—and tested across 15 diverse image classification datasets demonstrate that POMONAG outperforms existing state-of-the-art methods, including DiffusionNAG. The generated architectures achieve superior accuracy while satisfying various computational constraints and require significantly fewer trained models, highlighting the efficiency of the proposed technique.

The key contributions of this work include:

- **Diffusion-based Many-Objective Optimisation.** POMONAG introduces a many-objective diffusion approach within Transferable NAS, optimising neural architectures for accuracy, number of parameters, MACs, and inference latency.

- **Pareto-Optimal Architecture Generation.** By computing Pareto Fronts, POMONAG effectively navigates trade-offs among multiple objectives, generating architectures that offer balanced compromises suitable for diverse deployment scenarios.

- **Enhanced Meta-Datasets.** New Meta-Datasets have been developed to improve the Performance Predictors' ability to predict architecture performance across various tasks and datasets.

- **Refined Performance Predictor Models.** The Performance Predictor models have been enhanced to increase prediction accuracy and effectiveness in guiding the architecture generation process.

- **State-of-the-Art Transferable NAS Model.** POMONAG establishes a new benchmark by generating architectures that are high-performing and adaptable to various computational constraints.

The POMONAG model's code and the accompanying Meta-Datasets will be made publicly available upon publication of this paper at the following link.

## 2 RELATED WORKS

**Neural Architecture Search.** Neural Architecture Search (NAS) seeks to automate neural architecture design, removing the need for manual trial-and-error processes. Early methods, such as reinforcement learning (Zoph & Le, 2016; Zoph et al., 2018), evolutionary algorithms (Real et al., 2019; Lu et al., 2019), and gradient-based techniques (Xie et al., 2018; Dong & Yang, 2019b), are computationally expensive as they require full training of numerous architectures.

**One-shot NAS.** To mitigate computational costs, one-shot NAS methods employ weight sharing among candidate architectures. ENAS (Pham et al., 2018) uses an RNN controller to generate sub-networks; DARTS (Liu et al., 2018) relaxes the search space into a continuous one; OFA (Cai et al., 2020) trains a single large network with many architectural choices, and OFAv2 (Sarti et al., 2023a) extends this approach with an enriched search space. While these methods reduce costs, they may face challenges in optimisation bias, training stability, or performance across diverse sub-networks.

**Multi- and Many-Objective NAS.** As application complexity grows, NAS methods increasingly consider multiple objectives. Approaches like MONAS (Hsu et al., 2018) and DPP-Net (Dong et al.,

2018) balance criteria such as accuracy, latency, and energy consumption. NSGANetV2 (Lu et al., 2020) handles up to 12 objectives. Methods such as POPNASv3 (Falanti et al., 2023), NAT (Lu et al., 2021), and NATv2 (Sarti et al., 2023b) offer Pareto-optimal solutions, addressing scalability and computational challenges inherent in multi-objective optimisation.

**BO-based NAS.** To further reduce computational costs, Bayesian Optimisation (BO) predictor-based methods (Luo et al., 2018; Yu et al., 2020) employ surrogate models to estimate architecture performance without full training. BANANAS (White et al., 2021) uses an ensemble of neural networks, while NASBOWL (Ru et al., 2021) combines graph kernels with Gaussian processes. However, these methods often play a passive role, mainly filtering architectures generated by other strategies.

**Transferable NAS.** Transferable NAS methods leverage knowledge from previous tasks to accelerate searches on new datasets. MetaD2A (Lee et al., 2021) employs meta-learning to generate dataset-specific architectures. TNAS (Shala et al., 2023) enhances predictor adaptability to unseen datasets using BO with a deep-kernel Gaussian process. While promising in reducing search times, these methods may still face inefficiencies in exploring the architecture space.

**Diffusion Models.** Diffusion models (Ho et al., 2020; Rombach et al., 2022) have shown outstanding generative performance by learning to reverse the process of gradually adding noise to data. In graph generation, GDSS (Jo et al., 2022) applies diffusion models to undirected graphs. However, their application to neural architecture generation, involving directed acyclic graphs with specific constraints, remains unexplored.

**DiffusionNAG.** DiffusionNAG introduces a conditional Neural Architecture Generation framework based on diffusion models (An et al., 2024), addressing inefficiencies in traditional NAS methods that require sampling and training numerous irrelevant architectures. It models neural architectures as directed graphs and employs a graph diffusion process for their generation. The forward diffusion process perturbs architectures with Gaussian noise, mapping the architecture distribution to a known prior. The reverse process then refines noise into valid architectures using a learned score function. A key component is a specialised Score Network designed to capture dependencies between nodes, reflecting the computational flow in directed acyclic graphs. This network utilises Transformer blocks with an attention mask and incorporates positional embeddings to accurately represent the topological ordering of layers, ensuring that generated architectures adhere to specific search space rules. DiffusionNAG integrates a parameterised predictor into the reverse diffusion process, forming a predictor-guided conditional generation scheme. This allows the model to generate task-optimal architectures by sampling from regions more likely to satisfy desired properties.

In Transferable NAS scenarios, DiffusionNAG employs a meta-learned dataset-aware predictor conditioned on image classification datasets. This predictor, trained over a distribution of tasks, enables accurate predictions for unseen datasets without additional training. By integrating this predictor into the conditional generative process, DiffusionNAG facilitates efficient architecture generation for new tasks. By leveraging predictor-guided conditional architecture generation within the diffusion framework, DiffusionNAG reduces computational overhead and enhances search efficiency in NAS.

# 3 METHOD

This section outlines the techniques developed and implemented in POMONAG, a Transferable NAS model derived from the DiffusionNAG framework proposed by An et al. (2024). It presents the Many-Objective Reverse Diffusion Guidance process, the creation of new Meta-Datasets, and enhancements to the Performance Predictors. Additionally, it explains the Pareto Front Stretching and Pareto Front Filtering techniques used to optimise and refine the architecture generation process.

## 3.1 MANY-OBJECTIVE REVERSE DIFFUSION GUIDANCE

The diffusion model in DiffusionNAG employs the Reverse Diffusion Process for architecture generation, described as:

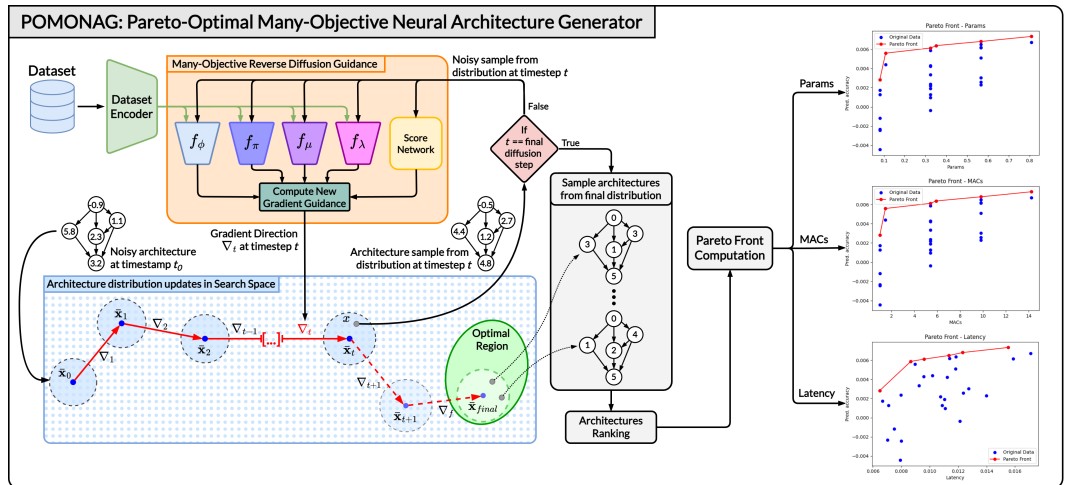

Figure 1: Overview of the POMONAG framework illustrating the Many-Objective Reverse Diffusion Guidance process.

$$dA_t = \left[ f_t(A_t) - g_t^2 \nabla_{A_t} \log p_t(A_t) \right] d\bar{t} + g_t d\bar{w}.$$

The score function $\nabla_{A_t} \log p_t(A_t)$ is approximated by introducing the **Score Network**, which iteratively applies the transformation $s_\theta(A_t, t)$ to the noisy architecture $A_t$. The process is guided using the dataset-aware **Performance Predictor** $f_\phi(y|\tilde{D}, A_t)$, resulting in:

$$dA_t = \left\{ f_t(A_t) - g_t^2 \left[ \boxed{s_\theta(A_t, t)} + k_\phi \nabla_{A_t} \log \boxed{f_\phi(y|\tilde{D}, A_t)} \right] \right\} d\bar{t} + g_t d\bar{w}.$$

The **Score Network** is responsible for denoising the noisy architecture sampled from the generative distribution at each denoising step. The output of the **Performance Predictor** is used to compute the guidance term that steers the generation process towards regions of the search space with a higher density of accurate architectures for the given dataset.

In POMONAG's Reverse Diffusion Guidance process, the generation gradient is directed in a many-objective fashion by simultaneously considering three additional terms:

- $k_\pi \nabla_{A_t} \log f_\pi(p|\tilde{D}, A_t)$, where $p$ represents the number of parameters of architecture $A_t$ and $k_\pi$ is a constant scaling factor;

- $k_\mu \nabla_{A_t} \log f_\mu(m|\tilde{D}, A_t)$, where $m$ represents the number of MACs of architecture $A_t$ and $k_\mu$ is a constant scaling factor;

- $k_\lambda \nabla_{A_t} \log f_\lambda(l|\tilde{D}, A_t)$, where $l$ represents the inference latency of architecture $A_t$ and $k_\lambda$ is a constant scaling factor.

Each term represents the log-likelihood of architecture $A_t$ satisfying the corresponding metric evaluated on dataset $D$ at timestamp $t$. These three terms are approximated by introducing three specialised Performance Predictors. The formulation for the new Many-Objective Reverse Diffusion Guidance is defined as:

$$dA_t = \left\{ f_t(A_t) - g_t^2 \left[ \boxed{s_\theta(A_t, t)} + k_\phi \nabla_{A_t} \log \boxed{f_\phi(y|\tilde{D}, A_t)} \right. \right.$$
$$\left. \left. + k_\pi \nabla_{A_t} \log \boxed{f_\pi(p|\tilde{D}, A_t)} + k_\mu \nabla_{A_t} \log \boxed{f_\mu(m|\tilde{D}, A_t)} + k_\lambda \nabla_{A_t} \log \boxed{f_\lambda(l|\tilde{D}, A_t)} \right] \right\} d\bar{t} + g_t d\bar{w}.$$

Each Performance Predictor – the Accuracy Predictor $f_\phi$, Parameters Predictor $f_\pi$, MACs Predictor $f_\mu$, and Inference Latency Predictor $f_\lambda$ – is trained to predict the corresponding metric of architecture $A_t$ when presented with dataset $D$. Similar to the Accuracy Predictor $f_\phi$, the outputs of each additional Performance Predictor are used to adjust the unconditioned gradient direction and update the generative probability distribution at each

denoising step during the diffusion phase. In doing so, the generative probability distribution is progressively guided towards regions of the search space that yield optimal architectures in terms of **accuracy**, **number of parameters**, number of operations (**MACs**), and **inference latency**. The entire many-objective generation process of POMONAG is depicted in Figure 1.

## 3.2 META-DATASET

The training of the Score Network, as well as the Performance Predictors, requires a Meta-Dataset containing the types of architecture structures to be generated, along with their characteristics and performance metrics. These architectures must be sampled from a sufficiently large search space and subsequently trained to solve specific tasks using a broad pool of datasets. This approach generalises the models and enables the Score Network to perform dataset-aware generation.

In line with practices used in other models within the Transferable NAS family (Lee et al., 2021; Shala et al., 2023; An et al., 2024), the datasets used for training the architectures – whose metadata will compose the Meta-Dataset – are extracted from the ImageNet32 dataset (Chrabaszcz et al., 2017). For each iteration, extraction is performed randomly by selecting 20 classes from ImageNet32. All corresponding samples of these classes form the dataset $D$. In parallel, an architecture $A$ is selected from the designated search space to be associated with this classification task. The structure of this architecture is stored in encoded form. For more information regarding the encoding of architectures, refer to Appendix A. Subsequently—and in contrast to other approaches—the number of parameters, MACs, and the inference latency on a single $32 \times 32$ sample are calculated for the architecture. Since latency is highly susceptible to noise, each measurement is repeated 100 times; measurements outside the 90% confidence interval are discarded, and the mean of the remaining values is computed.

Finally, the architecture is trained on the extracted dataset, split into training, validation, and test sets (80/10/10), and its accuracy is calculated on the test split. This information is aggregated and stored as a new tuple in the Meta-Dataset, forming a triplet:

- **Dataset**: The dataset $D$ of 20 classes extracted from ImageNet32.
- **Architecture**: The structure of the architecture $A$ in encoded form, extracted from the designated search space.
- **Objectives**: The test accuracy, parameters, MACs, and latency values of architecture $A$ on dataset $D$.

The search spaces used in this work are NASBench201 (Dong & Yang, 2020) and MobileNetV3 (Howard et al., 2019), utilised independently to create two distinct Meta-Datasets. The choice of training and augmentation techniques is as crucial as the size of the Meta-Datasets to obtain accurate Performance Predictors and generate high-performing and efficient architectures. To this end, the Meta-Datasets created for training the models composing POMONAG have undergone the following modifications compared to those used in the DiffusionNAG model:

- **Training Pipeline Optimisation.** The training pipelines for the architectures included in the Meta-Datasets have been optimised to maximise accuracy performance through hyperparameter tuning. For each search space, five architectures were selected and five associated datasets were extracted, as previously described. For each architecture, 50 optimisation steps were executed using Optuna (Akiba et al., 2019), incorporating a Tree-structured Parzen Estimator sampling method (Bergstra et al., 2011) alongside a Hyperband pruning mechanism (Li et al., 2018), aiming to maximise validation accuracy. For more information on the search space and the obtained pipelines, refer to Appendix D.
- **Dataset Expansion.** The size of the Meta-Datasets was increased by including a larger number of architecture-dataset pairs. For the NASBench201 search space, the cardinality was expanded from 4,630 to 10,000 triplets. Since the cardinality of the MobileNetV3-based Meta-Dataset was not provided by An et al. (2024), and given the considerable size of the search space, the number of triplets was set to 20,000.

## 3.3 SCORE NETWORK AND PERFORMANCE PREDICTORS

**Score Network.** The POMONAG framework adapts the core architecture of the Score Network from DiffusionNAG (An et al., 2024), incorporating modifications to accommodate the novel encodings detailed in Appendix A. The Score Network is responsible for iteratively denoising the encodings of the architectures to be generated. The fundamental structure remains consistent with DiffusionNAG. The input embeddings combine operation information ($\text{Emb}_{\text{ops}}$), node positions ($\text{Emb}_{\text{pos}}$), and time step ($\text{Emb}_{\text{time}}$):

$$\text{Emb}_i = \text{Emb}_{\text{ops}(v_i)} + \text{Emb}_{\text{pos}(v_i)} + \text{Emb}_{\text{time}(t)}$$

where $v_i$ represents the $i$-th row of the operator type matrix $V$. The training strategy for the Score Network remains consistent with that defined by An et al. (2024).

**Performance Predictors.** The Performance Predictors are models designed to predict the characteristics of architectures during and after the generation phase. In the first instance, the objective is to calculate the regression error used as Reverse Diffusion Guidance; in the second, the aim is to estimate the performance of the architecture once denoised, allowing for ranking without the need for any training and thereby identifying the most promising architectures.

In DiffusionNAG, there are two Performance Predictors: one for estimating the accuracy of noisy architectures during generation, and one for estimating the accuracy of denoised architectures. In POMONAG, there are five Performance Predictors. Four are dedicated to the respective estimation of accuracy, parameters, MACs, and inference latency of noisy architectures during the diffusion phase. The fifth estimates the accuracy of denoised architectures, given that the other metrics can be extracted with negligible overhead.

The Performance Predictors in POMONAG retain a similar architecture to that in DiffusionNAG. The substantial modifications pertain to the training procedure – which has been revised to maximise the Spearman correlation between the predicted values and the actual metrics – the size and content of the Meta-Dataset used during training, and the model employed as the Dataset Encoder. Specifically, while DiffusionNAG employs a ResNet18 architecture, POMONAG utilises a Vision Transformer (ViT-B-16) model for dataset embedding. From the conducted ablation studies, where the two Performance Predictors of DiffusionNAG are gradually adapted to the version used in POMONAG, a gradual improvement in terms of Spearman correlation is observable. Concretely, this amounts to improvements of +0.168 and +0.117 for the versions applied to noisy and denoised architectures, respectively. More details on the study are presented in Appendix C.

## 3.4 PARETO FRONT FILTERING AND STRETCHING

In DiffusionNAG, architectures are generated in batches of size 256. The scaling factor $k_\phi$ used to weight the contribution of the Performance Predictor during the generation process is set to 10000 and remains constant across all experiments. The generated architectures are then validated by filtering out those with inadequate or incorrect structures. Subsequently, the Performance Predictor for denoised architectures estimates their accuracy, obviating the need to train them. The architectures are then ranked based on this estimate, and the top five are returned as the output of the generation.

**Pareto Front Filtering.** In POMONAG, an additional filtering step is introduced through the construction of a Pareto Front, aimed at leveraging the secondary metrics. Specifically, after the many-objective generation of architectures, the elimination of invalid configurations, and the estimation of accuracies via the Performance Predictor, a Pareto Front is constructed for each of the three secondary metrics, and only the dominant architectures are retained. This approach enables the selection of architectures that are Pareto-optimal, allowing for choices based on trade-offs between accuracy and secondary metrics. To this end, three configurations extractable from each Pareto Front are identified. The configuration POMONAGAcc represents the architecture for which the highest accuracy is predicted; the configuration POMONAGBal represents the architecture for which the ratio between predicted accuracy and the considered secondary metric is highest; finally, the configuration POMONAG$_{\text{Eff}}$ represents the architecture with the lowest value of the secondary metric and therefore the most efficient.

**Pareto Front Stratching.** The Many-Objective Reverse Diffusion Guidance process introduced in POMONAG utilises four different guides, one for each Performance Predictor, to model the distribution of the architecture space. To optimise the corresponding scaling factors, POMONAG employs a dynamic approach rather than a fixed value as in DiffusionNAG. This optimisation process aims to maximise the mean of the estimated accuracies of the architectures in the Pareto Front by adjusting the scaling factors corresponding to each guide.

This optimisation leverages Optuna (Akiba et al., 2019), incorporating a Tree-structured Parzen Estimator sampling method (Bergstra et al., 2011) alongside a Hyperband pruning mechanism (Li et al., 2018). The process involves 100 evaluations on four datasets (CIFAR10, CIFAR100, Aircraft, Oxford III Pets). This procedure is replicated for each search space.

To fully exploit the secondary metrics, POMONAG introduces a novel technique termed Pareto Front Stretching. This entails generating architectures in two phases, each with batches of 128 elements. The first batch favours efficient architectures with respect to the secondary metrics, assigning greater weight to the predictors of parameters, MACs, and inference latency. The second batch aims to obtain highly accurate architectures while still maintaining constraints on their complexity.

To implement these two generation processes, the optimisation of the scaling factors is executed twice with different constraints. For efficient architectures, the scaling factor for the accuracy guide is searched within the interval $[1000, 5000]$, while for the other guides it is within $[100, 500]$. For highly accurate architectures, the

Table 1: Comparison with Transferable NAS methods on the MobileNetV3 search space. The POMONAG architectures in this comparison correspond to the POMONAG$_{Acc}$ configuration. The best scores are highlighted in **bold**.

| Dataset | Stats. | MetaD2A Lee et al. (2021) | TNAS Shala et al. (2023) | DiffusionNAG An et al. (2024) | POMONAG (ours) |
|---------|--------|---------|------|-------------|---------|
| CIFAR10 | Max | 97.45±0.07 | 97.48±0.14 | 97.52±0.07 | **97.85±0.14** |
|         | Mean | 97.28±0.01 | 97.22±0.00 | 97.39±0.01 | **97.85±0.14** |
|         | Min | 97.09±0.13 | 95.26±0.09 | 97.23±0.06 | **97.85±0.14** |
| CIFAR100 | Max | 86.00±0.19 | 85.95±0.29 | 86.07±0.16 | **86.18±0.11** |
|          | Mean | 85.56±0.02 | 85.30±0.04 | 85.74±0.04 | **86.18±0.11** |
|          | Min | 84.74±0.13 | 81.30±0.18 | 85.42±0.08 | **86.18±0.11** |
| Aircraft | Max | 82.18±0.70 | 82.31±1.31 | 82.28±0.29 | **87.62±0.12** |
|          | Mean | 81.19±0.11 | 80.86±0.15 | 81.47±0.05 | **87.62±0.12** |
|          | Min | 79.71±0.54 | 74.99±6.65 | 80.88±0.54 | **87.62±0.12** |
| Oxford III Pets | Max | 95.28±0.50 | 95.04±0.44 | **95.34±0.29** | 95.28±0.09 |
|                 | Mean | 94.55±0.03 | 94.47±0.10 | 94.75±0.14 | **95.28±0.09** |
|                 | Min | 93.68±0.16 | 92.39±0.47 | 94.28±0.17 | **95.28±0.09** |

scaling factor for the accuracy guide is optimised within $[10,000, 50,000]$, while for the other metrics within $[10, 50]$.

The optimisation results yielded specific values for each search space. For NASBench201, the optimal scaling factors for efficient architectures were 4732, 482, 421, and 368, respectively for accuracy, parameters, MACs, and latency. For highly accurate architectures, the values are 24,943, 12, 26, and 13. In the case of MobileNetV3, for efficient architectures, the values obtained are 4987, 494, 478, and 481, while for highly accurate ones, 48,321, 21, 16, and 39.

This approach allows for a more thorough exploration of the solution space, generating an overall set of architectures more widely spread with respect to the secondary metrics. The effect is therefore a diversified generation of architectures and a Pareto Front with a more elongated shape. For more information, the reader is referred to Appendix B.

## 4 EXPERIMENTS AND RESULTS

This section presents the experiments conducted and the results obtained to evaluate the performance of POMONAG compared to other NAS models, particularly DiffusionNAG. All experiments were conducted using an NVIDIA Quadro RTX 6000 graphics card.

**Transferable NAS Evaluation on MobileNetV3.** In Table 1, the comparison between POMONAG and other Transferable NAS methods on the MobileNetV3 search space is presented. POMONAG achieves the best performance in terms of minimum, mean, and maximum accuracy on almost all the datasets considered. Notably, on the Aircraft dataset, POMONAG reaches an accuracy of 87.62%, significantly surpassing other methods. On CIFAR10, CIFAR100, and Oxford III Pets, POMONAG also demonstrates superior or comparable performance to the state of the art. Unlike other approaches, where the generation of architectures yields multiple alternatives and diversifies the obtained performances, by exploiting the Pareto Front and selecting the POMONAG$_{Acc}$ configuration, it is possible to reduce the number of architectures to be trained to just one. These results demonstrate POMONAG's ability to generate architectures that, even when considering multiple conditionings based on competing metrics, manage to outperform previous state-of-the-art methods.

**NAS Evaluation on NASBench201.** Table 2 compares the performance of POMONAG with that of other NAS techniques on the NASBench201 search space across the datasets CIFAR10, CIFAR100, Aircraft, and Oxford III Pets. POMONAG achieves the best accuracies on all datasets, using only one trained architecture per dataset, mirroring the approach taken for the MobileNetV3 search space. Specifically, for the CIFAR10 and CIFAR100 datasets, where all other Transferable NAS methods obtain accuracy results with a 95% confidence interval of 0.00 due to lookup procedures, POMONAG presents a non-zero value as its performances were recalculated over three different runs, confirming superior performance. It is crucial to note that POMONAG minimises the number of architectures to be trained to only the final architecture, thereby reducing computational complexity to a minimum.

**Performance Predictors.** Optimising the Performance Predictors is fundamental for effectively guiding the many-objective conditional generation process. The comparative analysis summarised in Table 3 highlights

Table 2: Comparison of NAS techniques on the NASBench201 search space across four datasets. 'Trained Archs' indicates the number of neural architectures trained to achieve the reported accuracy. Results show mean accuracy $\pm$ 95% confidence intervals over three runs. The POMONAG architectures used in this comparison correspond to the POMONAG$_{Acc}$ configuration. Best scores are highlighted in **bold**.

| Type | Method | CIFAR10 | | CIFAR100 | | Aircraft | | Oxford III Pets | |
|---|---|---|---|---|---|---|---|---|---|
| | | Accuracy ↑ | Trained Archs ↓ | Accuracy ↑ | Trained Archs ↓ | Accuracy ↑ | Trained Archs ↓ | Accuracy ↑ | Trained Archs ↓ |
| One-shot NAS | RSPS Li & Talwalkar (2020) | 84.07±3.61 | N/A | 52.31±5.77 | N/A | 42.19±3.88 | N/A | 22.91±1.65 | N/A |
| | SETN Dong & Yang (2019a) | 87.64±0.00 | N/A | 59.09±0.24 | N/A | 44.84±3.96 | N/A | 25.17±1.68 | N/A |
| | GDAS Dong & Yang (2019b) | 93.61±0.09 | N/A | 70.70±0.30 | N/A | 53.52±0.48 | N/A | 24.02±2.75 | N/A |
| | PC-DARTS Xu et al. (2020) | 93.66±0.17 | N/A | 66.64±2.34 | N/A | 26.33±3.40 | N/A | 25.31±1.38 | N/A |
| | DrNAS Chen et al. (2021) | 94.36±0.00 | N/A | 73.51±0.00 | N/A | 46.08±7.00 | N/A | 26.73±2.61 | N/A |
| BO-based NAS | BOHB Falkner et al. (2018) | 93.61±0.52 | >500 | 70.85±1.28 | >500 | - | - | - | - |
| | GP-UCB Srinivas et al. (2012) | 94.37±0.00 | 58 | 73.14±0.00 | 100 | 41.72±0.00 | 40 | 40.60±1.10 | 11 |
| | BANANAS White et al. (2021) | 94.37±0.00 | 46 | 73.51±0.00 | 88 | 41.72±0.00 | 40 | 40.15±1.59 | 17 |
| | NASBOWL Ru et al. (2021) | 94.34±0.00 | 100 | 73.51±0.00 | 87 | 53.73±0.83 | 40 | 41.29±1.10 | 17 |
| | HEBO Cowen-Rivers et al. (2022) | 94.34±0.00 | 100 | 72.62±0.20 | 100 | 49.32±6.10 | 40 | 40.55±1.15 | 18 |
| Transferable NAS | TNAS Shala et al. (2023) | 94.37±0.00 | 29 | 73.51±0.00 | 59 | 59.15±0.58 | 26 | 40.00±0.00 | 6 |
| | MetaD2A Lee et al. (2021) | 94.37±0.00 | 100 | 73.34±0.04 | 100 | 57.71±0.20 | 40 | 39.04±0.20 | 40 |
| | DiffusionNAG An et al. (2024) | 94.37±0.00 | 5 | 73.51±0.00 | 5 | 59.63±0.92 | 2 | 41.32±0.84 | 2 |
| | POMONAG (Ours) | **95.42**±0.12 | **1** | **75.94**±0.24 | **1** | **63.38**±0.51 | **1** | **68.82**±0.19 | **1** |

Table 3: Comparison of Spearman's correlation coefficients for Performance Predictors between DiffusionNAG and POMONAG on the NASBench201 and MobileNetV3 search spaces. Higher coefficients indicate better predictions. 'Accuracy', 'Params', 'MACs', and 'Latency' refer to predictions for noisy architectures during diffusion, while 'Accuracy*' represents predictions for denoised architectures after diffusion. 'N/A' indicates experiments that could not be replicated; '-' indicates metrics not applicable to DiffusionNAG. Best scores are highlighted in **bold**.

| Method | NASBench201 | | | | | MobileNetV3 | | | | |
|---|---|---|---|---|---|---|---|---|---|---|
| | Accuracy | Accuracy* | Params | MACs | Latency | Accuracy | Accuracy* | Params | MACs | Latency |
| DiffusionNAG | 0.687 | 0.767 | - | - | - | N/A | N/A | - | - | - |
| POMONAG | **0.855** | **0.884** | **0.666** | **0.656** | **0.470** | **0.857** | **0.988** | **0.828** | **0.847** | **0.618** |

a marked superiority of the predictors employed in POMONAG compared to those of DiffusionNAG. In the context of NASBench201, a substantial increase in Spearman's correlation for accuracy is observed: from 0.687 to 0.855 for noisy architectures and from 0.767 to 0.884 for denoised ones. POMONAG also stands out for introducing efficient predictors for parameters, MACs, and latency, absent in DiffusionNAG. These new predictors show significant correlations—0.666 for parameters, 0.656 for MACs, and 0.470 for latency – thus providing reliable guidance for many-objective optimisation. In the MobileNetV3 search space, POMONAG achieves even higher correlations, exceeding 0.8 for accuracy, parameters, and MACs. Appendix C presents a detailed analysis of the ablation studies related to these predictors, corroborating the effectiveness of the implemented modifications.

**Generation Performance.** The comparative analysis in Table 4 shows generation metrics between Diffusion-NAG and POMONAG. Validity measures the percentage of generated architectures that adhere to the constraints of the search space; uniqueness assesses the diversity among the generated architectures; and novelty indicates the proportion of architectures not present in the training dataset. In the NASBench201 search space, POMONAG demonstrates significant improvements across all metrics, particularly in uniqueness (34.14% vs. 5.10%) and novelty (37.41% vs. 19.87%) compared to DiffusionNAG. For the MobileNetV3 search space, a slight decrease in validity is observed for POMONAG (72.58% vs. 99.09%), likely due to a more ambitious exploration of the search space driven by many-objective conditioning. Nonetheless, POMONAG maintains high levels of uniqueness (91.79%) and novelty (100%), illustrating its capability to generate diverse and innovative architectures while simultaneously achieving superior classification performance.

**Many-Objective Evaluation.** A comprehensive comparison between DiffusionNAG and the three variants of POMONAG (**Efficient**, **Balanced**, **Accurate**) was conducted across both NASBench201 and MobileNetV3 search spaces considering all the key metrics involved in the many-objective optimisation – accuracy, number of parameters, MACs, and inference latency. The results, summarised in Table 5, represent the means over 15 diverse image classification datasets, depicted in more details in Appendix B.

Table 4: Comparison of generation metrics—'Validity', 'Uniqueness', and 'Novelty'—between DiffusionNAG and POMONAG on the NASBench201 and MobileNetV3 search spaces. Results show mean values ± 95% confidence intervals over three runs. Best scores are highlighted in **bold**.

| Method | NASBench201 | | | MobileNetV3 | | |
|---|---|---|---|---|---|---|
| | Validity ↑ | Uniqueness ↑ | Novelty ↑ | Validity ↑ | Uniqueness ↑ | Novelty ↑ |
| DiffusionNAG | 98.97±0.29 | 5.10±0.33 | 19.87±1.35 | **99.09±0.28** | **100.00±0.00** | **100.00±0.00** |
| POMONAG (ours) | **99.97±0.10** | **34.14±17.33** | **37.41±7.73** | 72.58±5.84 | 91.79±7.32 | **100.00±0.00** |

Table 5: Comparative analysis of DiffusionNAG and POMONAG variants on the NASBench201 and MobileNetV3 search spaces. Metrics include accuracy, number of parameters (millions), MACs (millions), and latency (milliseconds). Values are means over the 15 image classification datasets considered, with ± 95% confidence intervals. **Bold** indicates the best performance, and underlined indicates the second-best. Green and red shading denote POMONAG's improvements or deteriorations, respectively, relative to DiffusionNAG.

| Model | NASBench201 | | | | MobileNetV3 | | | |
|---|---|---|---|---|---|---|---|---|
| | Accuracy ↑ | Params [M] ↓ | MACs [M] ↓ | Latency [ms] ↓ | Accuracy ↑ | Params [M] ↓ | MACs [M] ↓ | Latency [ms] ↓ |
| DiffusionNAG | 77.83±10.86 | 1.02±0.01 | 18.02±0.21 | 18.83±0.46 | 87.29±6.32 | 7.86±1.05 | 100.75±8.18 | 32.13±2.95 |
| POMONAG$_{Eff}$ | 70.25±12.46 | **0.10±0.03** | **1.27±0.63** | **6.51±0.77** | 88.66±5.28 | **3.47±0.01** | **33.69±0.85** | **17.96±0.76** |
| POMONAG$_{Bal}$ | 77.91 ± 10.69 | 0.43 ± 0.18 | 7.33 ± 3.21 | 11.06 ± 2.12 | 89.02 ± 5.22 | 4.24 ± 0.48 | 38.02 ± 6.92 | 21.62 ± 3.14 |
| POMONAG$_{Acc}$ | **81.89 ± 9.21** | 0.65 ± 0.11 | 11.40 ± 2.06 | 13.68 ± 1.17 | **89.96 ± 4.86** | 6.01 ± 0.37 | 90.92 ± 4.62 | 34.24 ± 2.19 |

In the NASBench201 search space, POMONAGAcc achieves the highest average accuracy (81.89%), surpassing DiffusionNAG's 77.83%, while also improving upon all secondary metrics. This indicates that accuracy gains do not come at the expense of efficiency. The POMONAGEff variant achieves remarkable reductions in parameters (-90%), MACs (-93%), and latency (-65%) compared to DiffusionNAG, albeit with a trade-off in accuracy. This highlights POMONAG's capability to generate highly efficient architectures suitable for resource-constrained environments. The POMONAG$_{Bal}$ configuration offers a balanced compromise, maintaining an accuracy comparable to DiffusionNAG while significantly enhancing efficiency metrics. In the MobileNetV3 search space, all POMONAG variants outperform DiffusionNAG in terms of average accuracy. Notably, POMONAGEff achieves significant reductions in parameters (-56%), MACs (-67%), and latency (-44%), demonstrating its effectiveness in generating compact and fast architectures suitable for devices with limited computational capabilities. The POMONAGAcc variant attains the highest average accuracy (89.96%), with a 24% reduction in parameters and a 10% reduction in MACs compared to DiffusionNAG. This illustrates that it is possible to enhance performance while also improving efficiency. These results underscore POMONAG's effectiveness in generating Pareto-optimal architectures that provide a balanced trade-off between accuracy and efficiency, catering to various deployment scenarios and resource constraints.

**Generation and Training Time.** The time required for generating and training the architectures depends on the search space, the selected POMONAG configuration, and the size of the dataset used for training. For the generation phase, the diffusion process takes an average of approximately 5 minutes and 45 seconds (±32 seconds) on NASBench201, and about 18 minutes and 15 seconds (±1 minute and 12 seconds) on MobileNetV3. Regarding training times, utilising the optimised pipelines detailed in Appendix D, the training durations for NASBench201 range from approximately 2 hours and 33 minutes (±54 minutes) to 3 hours and 32 minutes (±1 hour and 19 minutes), depending on the POMONAG configuration. For MobileNetV3, training times vary from about 2 hours and 8 minutes (±31 minutes) to 2 hours and 26 minutes (±39 minutes), again contingent on the configuration used. Considering that only a single generation and a single training run are required, and in light of the superior performances achieved in terms of both accuracy and efficiency, POMONAG represents a substantial advancement in the field of Neural Architecture Search. This reduction in computational time and resources not only accelerates the development process but also makes the approach more accessible for practical applications where time and computational budget are limited.

## 5 CONCLUSION AND FUTURE DIRECTIONS

This paper has presented POMONAG, an innovative framework for generating neural architectures optimised in a many-objective context. By integrating advanced techniques like Many-Objective Reverse Diffusion Guidance, the creation of extended Meta-Datasets, and the adoption of improved Performance Predictors, POMONAG addresses and overcomes limitations of existing approaches, effectively combining many-objective

and Transferable NAS paradigms. The conducted experiments demonstrate that POMONAG is capable of generating architectures offering an effective balance between accuracy and computational efficiency. Specifically, it achieves superior performance across various datasets and search spaces requiring training only the identified optimal architecture. As a future direction, extending POMONAG's approach to other computer vision tasks – such as segmentation and object detection – is envisaged. This trajectory aims to develop a foundational NAS model for computer vision tasks that is both task-aware and dataset-aware, capable of effectively adapting to a variety of applications, domains, and computational constraints. Such an advancement would further enhance the adaptability and applicability of NAS methodologies in diverse real-world scenarios.

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

## A    META-DATASETS

This appendix describes the search spaces utilised in this work, the encoding and sampling strategies employed for the creation of the Meta-Datasets, and introduces the Meta-Datasets themselves.

### A.1    NASBench201 Search Space

NASBench201, introduced by Dong & Yang (2020), is a standardised search space for Neural Architecture Search (NAS) tasks within the domain of image classification. This search space is confined to the composition of an optimal cell, which is used in series to construct the final architecture. A cell comprises four fixed nodes, representing the summation operation of the feature maps received as input, and six possible connections between the nodes. The connections are associated with operations for transforming and adapting the feature maps. The available operations are Zeroise, Skip connection, 1x1 2D Convolution, 3x3 2D Convolution, and 3x3 Average Pooling. Each convolution operation is followed by a Batch Normalisation layer and a ReLU activation. This leads to a total of $5^6 = 15,625$ unique architectures within the NASBench201 search space. The evaluation of models in NASBench201 is conducted through full training on three datasets: CIFAR10, CIFAR100, and ImageNet16. For each architecture, metrics such as accuracy, cross-entropy loss, training time, number of parameters, MACs, and inference latency are provided.

**Architecture Encoding and Sampling.** The encoding of NASBench201 architectures used in this work involves two complementary matrices: an operations matrix and an adjacency matrix, as proposed by Dong & Yang (2020).

The operations matrix describes the transformations applied to the feature maps between the nodes of the graph. This $8 \times 7$ matrix is structured such that each cell represents a possible edge of the directed acyclic graph. There are eight rows—one for the input, six for the possible intermediate connections, and one for the output—and seven columns, one for each possible operation, including placeholders for the input and output. The matrix is binary: a '1' indicates that the connection is assigned the corresponding operation, and a '0' otherwise. In this sense, each row has exactly one '1', since each connection must be assigned an operation. Parallelly, the adjacency matrix defines the connection structure of the graph. Also of size $8 \times 8$, this binary matrix is fixed for all architectures in NASBench201. The elements of the matrix are '1' to indicate the presence of a connection between two nodes summing the feature maps, and '0' to indicate the absence of a connection. Again, the first row and column represent the input, while the last row and column represent the output. The sampling of architectures is optimised by exploiting the knowledge of the evaluations present in NASBench201. Specifically, the top-250 most performant architectures are identified by calculating the average relative to the three datasets used by the authors for their evaluation. Sampling is conducted such that there is a 95% probability of sampling from this subset of high-performing architectures, and a 5% probability from the rest of the search space.

### A.2    MobileNetV3 Search Space

The MobileNetV3 search space (Howard et al., 2019), implemented via Once-for-All (OFA) introduced by Cai et al. (2020), is a flexible search space for NAS tasks in the domain of image classification on mobile devices. This search space is defined by a pre-trained super-network that supports various architectural configurations, allowing exploration of a wide range of sub-networks. The super-network comprises a sequence of inverted bottleneck blocks, similar to those in MobileNetV2 (Sandler et al., 2018). The main dimensions of the search space are:

- **Depth.** Each stage of the network can have a variable number of blocks, typically chosen from 2, 3, 4.
- **Expansion ratio.** The number of channels in each layer can be adjusted using two width multipliers.
- **Kernel size.** For depthwise convolutions, the kernel size can be $3 \times 3$, $5 \times 5$, or $7 \times 7$.
- **Input resolution.** The input image can have variable dimensions, typically from $128 \times 128$ to $224 \times 224$.

The input resolution is fixed at $224 \times 224$. Each of the five stages in the super-network, as provided by the original implementation, can include a Squeeze-and-Excitation module and use different activation functions (ReLU or h-swish) based on its position within the architecture. This approach leads to a search space comprising approximately $10^{19}$ unique architectures. The evaluation of models in this space is conducted through the selection of subgraphs of the pre-trained super-network, eliminating the need to train each architecture from scratch.

**Architecture Encoding and Sampling.** As with NASBench201, the encoding of MobileNetV3 architectures used in this work involves representations related to the operators of the architecture and the internal connections within the network. The operators are represented by a $21 \times 9$ matrix, with the first row dedicated to the

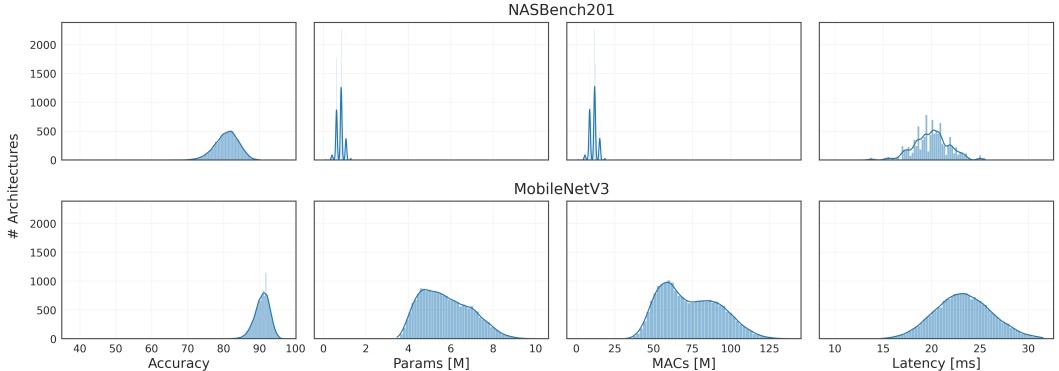

Figure 2: Distributions of key metrics in the proposed Meta-Datasets. Histograms display accuracy, number of parameters, MACs, and latency for architectures sampled from the NASBench201 (top row, 10,000 architectures) and MobileNetV3 (bottom row, 20,000 architectures) search spaces. The architectures were randomly selected and trained on subsets of ImageNet32, demonstrating the diversity within each search space.

placeholder of the width multiplier, followed by one row for each of the 20 blocks within the architecture. If the width multiplier is set to 1.0, then the first row is populated with zeros; if it is set to 1.2, the first row is populated exclusively with ones. From the second row onwards, the columns are divided into three groups of three, each representing a combination of expansion ratio and kernel size. The first group corresponds to an expansion ratio of 3, the second to 4, and the third to 6. Within each group, the three columns represent kernel sizes of 3×3, 5×5, and 7×7, respectively. Each row of the matrix contains a '1' in the position corresponding to the chosen combination of expansion ratio and kernel size for that block, and zeros elsewhere. If a block is inactive (as indicated by the depth mask), its corresponding row is filled with zeros. The adjacency matrix, on the other hand, represents the connections between the blocks of the architecture. It is a square matrix of size 20×20, where each element (i, j) is '1' if there is a direct connection from block $i$ to block $j$, and '0' otherwise. The structure of this matrix reflects the topology of the network, taking into account the variable depth of each stage. For this search space, the sampling of architectures is conducted uniformly with respect to the values of the width multiplier, depth, expansion ratio, and kernel size, drawing from a possible number of architectures equal to $2 \times 10^{19}$ (the factor of 2 arises from the addition of the width multiplier).

## A.3 META-DATASETS DISTRIBUTIONS

Figure 2 illustrates the distributions of key metrics for the Meta-Datasets created in this study. For the NAS-Bench201 search space, 10,000 architectures were sampled, while 20,000 were selected for the MobileNetV3 space. These architectures were sampled from their respective search spaces and then trained using the proposed pipelines on a dataset of 20 classes with 1,000 samples per class, randomly extracted from ImageNet32. The reported accuracy performance is based on a test set of 10 samples per class for the same 20 classes. The distributions shown are consistent with the characteristics of their respective search spaces. Notably, the graphs clearly demonstrate that MobileNetV3 architectures are, as expected, more complex yet relatively lightweight, and also more performant. This is due to their incorporation of more advanced architectural designs compared to NASBench201 architectures, as well as the benefit of fine-tuning during training.

# B    EXTENSIVE RESULTS

Figures 3 and 4 present a comprehensive comparison of neural architectures generated by DiffusionNAG and POMONAG across 15 diverse datasets: CIFAR10, CIFAR100, Aircraft, Oxford III Pets, BloodMNIST, DermaMNIST, EuroSAT, FashionMNIST, OCTMNIST, OrganAMNIST, OrganCMNIST, PathMNIST, STL10, TinyImageNet, and TissueMNIST. The comparisons are made using the NASBench201 and MobileNetV3 search spaces, respectively. The plots illustrate the trade-offs between accuracy and secondary metrics (parameters, MACs, and latency) for each dataset. POMONAG's results are represented by Pareto Fronts, showcasing the range of optimal solutions. Three key variants are highlighted: **Efficient**, emphasising minimal secondary metric; **Accurate**, prioritising highest predicted accuracy; and **Balanced**, optimising the ratio of predicted accuracy to secondary metric. DiffusionNAG's results, both reproduced (orange points) and as reported by An et al. (2024) (dashed lines), provide a benchmark for comparison. The 95% confidence intervals, averaged over three runs, underscore the robustness of the results. These visualisations demonstrate POMONAG's capability to generate a diverse set of architectures, effectively balancing performance and efficiency across various datasets and search spaces. The consistent outperformance of POMONAG, particularly in terms of Pareto-optimal solutions, highlights its effectiveness in navigating complex Neural Architecture Search spaces. It is important to note that the performances of POMONAG's architectures, as well as those replicated from DiffusionNAG and represented in both figures, were explicitly calculated and are not estimates from the Performance Predictors.

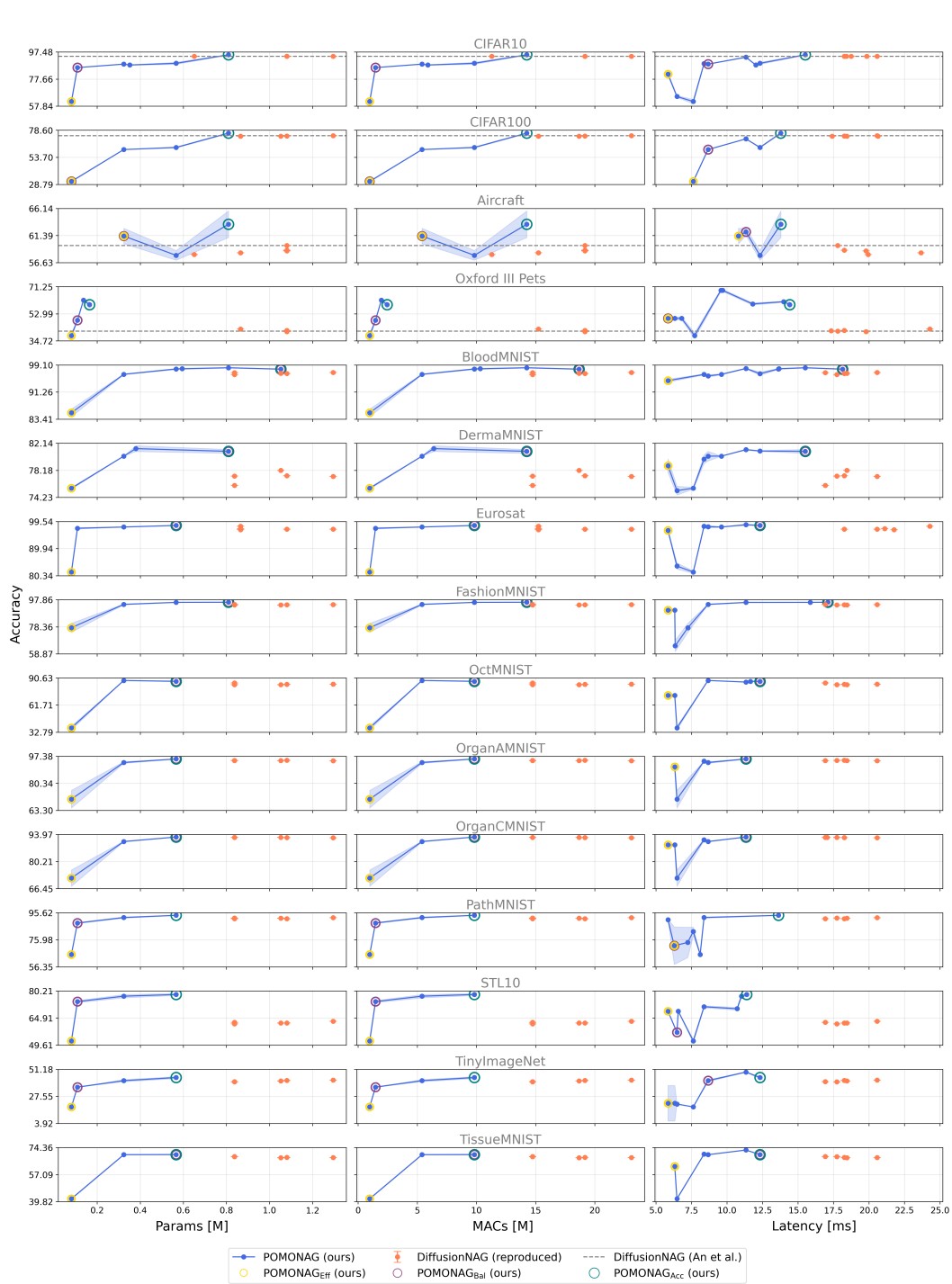

Figure 3: Performance comparison of architectures generated by DiffusionNAG and POMONAG on 15 datasets using the NASBench201 search space. Each plot illustrates accuracy versus a secondary metric (number of parameters, MACs, or latency). The blue lines represent POMONAG's Pareto Fronts, with circles highlighting the efficient (yellow), balanced (purple), and accurate (green) variants. Orange points depict DiffusionNAG results. Error bars show 95% confidence intervals averaged over three runs. Dashed lines indicate DiffusionNAG results as reported by An et al. (2024).

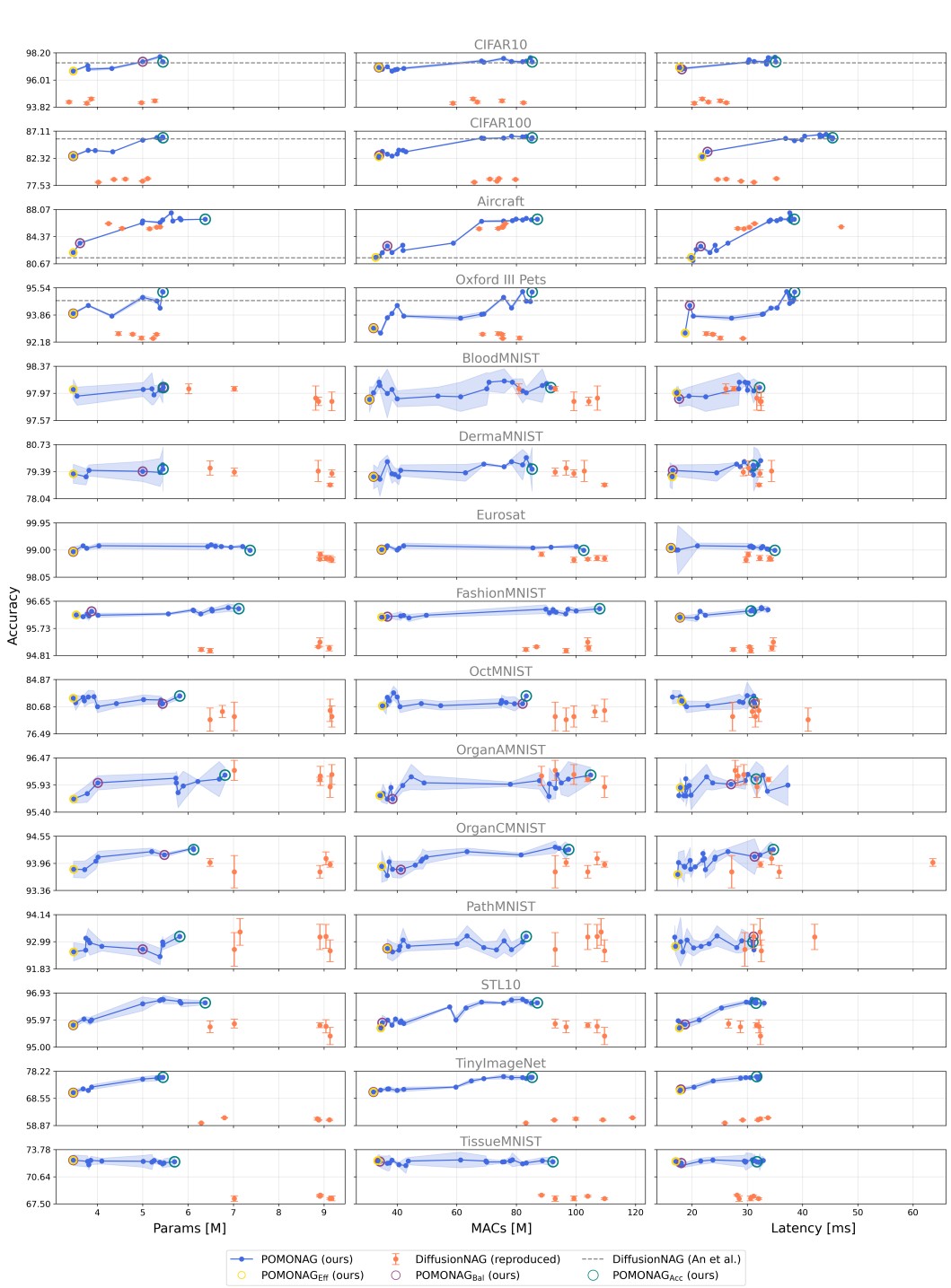

Figure 4: Performance comparison of architectures generated by DiffusionNAG and POMONAG on 15 datasets using the MobileNetV3 search space. Each plot illustrates accuracy versus a secondary metric (number of parameters, MACs, or latency). The blue lines represent POMONAG's Pareto Fronts, with circles highlighting the efficient (yellow), balanced (purple), and accurate (green) variants. Orange points depict DiffusionNAG results. Error bars show 95% confidence intervals averaged over three runs. Dashed lines indicate DiffusionNAG results as reported by An et al. (2024).

Table 6: Ablation study results for Performance Predictors on the NASBench201 search space. 'Performance Predictor' specifies the training and the encoding strategy used for the predictors. 'Archs Training' indicates the training pipeline for the architectures in the Meta-Dataset. 'Size' denotes the Meta-Dataset's cardinality. 'Accuracy' and 'Accuracy*' are Spearman correlations for noisy and denoised architectures, respectively. Best scores are highlighted in **bold**

| Performance Predictor | | Meta-Dataset | | Spearman Correlation | |
|---|---|---|---|---|---|
| Training Strategy | Dataset Encoder | Archs Training | Size | Accuracy | Accuracy* |
| An et al. (2024) | ResNet18 | An et al. (2024) | 4630 | 0.687 | 0.767 |
| Ours | ResNet18 | An et al. (2024) | 4630 | 0.705 | 0.772 |
| Ours | ResNet18 | Ours | 4630 | 0.822 | 0.854 |
| Ours | ResNet18 | Ours | 10000 | 0.842 | 0.884 |
| Ours | ViT-B-16 | Ours | 10000 | **0.855** | **0.884** |

## C   PERFORMANCE PREDICTORS ABLATION STUDY

An ablation study was conducted to evaluate the improvements introduced in the Performance Predictors on the NASBench201 search space. The study involved a series of modifications to the predictors and their training strategies, aiming to maximise the Spearman correlation between the predicted values and the actual metrics.

Initially, a new predictor was implemented using the same training pipeline as An et al. (2024), which yielded slight improvements in correlation scores over the baseline DiffusionNAG predictor. Specifically, the Spearman correlation for the accuracy of noisy architectures increased from 0.687 to 0.705, and for denoised architectures from 0.767 to 0.772. Subsequently, the introduction of a novel training strategy significantly enhanced performance for both noisy and denoised architecture accuracy estimates. This new training strategy involved switching from the Adam optimiser to AdamW, incorporating a cosine annealing learning rate scheduler, and introducing a weight decay of $5 \times 10^{-3}$. These modifications improved convergence and generalisation of the Performance Predictors, leading to Spearman correlations of 0.822 for noisy architectures and 0.854 for denoised architectures.

Expanding the Meta-Dataset size from 4,630 to 10,000 architectures further improved the results, with correlations reaching 0.842 for noisy architectures and 0.884 for denoised ones. This demonstrates the benefit of a larger and more diverse training set, providing the Performance Predictors with a richer array of examples to learn from. Finally, replacing the ResNet18 Dataset Encoder with a Vision Transformer (ViT-B-16) feature extractor led to the best performance, achieving Spearman correlations of 0.855 and 0.884 for noisy and denoised architectures, respectively. These results indicate that both the architectural choices and the training strategies for the Performance Predictors have a substantial impact on their effectiveness, directly influencing the quality of guidance provided during the diffusion process.

Table 7: Hyperparameter search space used for optimising the training pipelines for architectures in the Meta-Datasets.

| Hyperparameter | Search Space |
|---|---|
| Epochs | $\{50, 100, 200\}$ |
| Warm-up epochs | $\{-, 10, 20\}$ |
| Optimiser | $\{SGD, Adam, AdamW\}$ |
| Learning rate | $\{1\times10^{-1}, 1\times10^{-2}, 1\times10^{-3}, 1\times10^{-4}\}$ |
| Learning rate scheduler | $\{-, \text{Cosine annealing, Cosine annealing with restarts}\}$ |
| Weight decay | $\{-, 5\times10^{-3}, 5\times10^{-4}, 5\times10^{-5}\}$ |
| Label smoothing | $\{-, 0.1\}$ |
| Heavy augmentation | $\{-, \text{AutoAugment, TrivialAugmentWide, RandAugment, AugMix}\}$ |
| Random horizontal flip | $\{-, 0.5\}$ |
| Random erasing (Gaussian) | $\{-, 0.2\}$ |
| MixUp | $\{-, 0.2\}$ |
| CutMix | $\{-, 0.2\}$ |

Table 8: Optimal training hyperparameters identified for architectures in the NASBench201 and MobileNetV3 search spaces.

| Search Space | NASBench201 | MobileNetV3 |
|---|---|---|
| Weights initialisation | HeNormal initialisation | ImageNet-1k pre-training |
| Mixed precision | True | True |
| Training technique | Ordinary | Fine-tuning |
| Epochs | 200 | 50 |
| Warm-up epochs | 20 | - |
| Early stopping patience | 120 | 30 |
| | | |
| Optimiser | SGD | AdamW |
| Learning rate | $1\times10^{-1}$ | $1\times10^{-3}$ |
| Learning rate scheduler | Cosine annealing | - |
| Weight decay | $5\times10^{-4}$ | $5\times10^{-5}$ |
| Label smoothing | - | 0.1 |
| | | |
| Resizing | $32\times32$ | $224\times224$ |
| Heavy augmentation | TrivialAugmentWide | AugMix |
| Random horizontal flip | 0.5 | 0.5 |
| Padding (constant) | 4 | 21 |
| Random crop | $32\times32$ | $224\times224$ |
| Random erasing (Gaussian) | 0.2 | - |
| MixUp | - | 0.2 |

## D  HYPERPARAMETER TUNING AND OPTIMISATION

The training of the architectures included in the Meta-Datasets required meticulous hyperparameter tuning to maximise performance. For each search space – NASBench201 and MobileNetV3 – a comprehensive hyperparameter search was conducted to identify optimal training configurations. The hyperparameter search involved exploring a range of values for key training parameters, such as the number of epochs, warm-up epochs, optimisers, learning rates, learning rate schedulers, weight decay, label smoothing, and various data augmentation techniques. The search space for these hyperparameters is summarised in Table 7, providing an overview of the parameters considered during the optimisation process. The optimal training pipelines identified for each search space are detailed in Table 8.

