# OpenReview forum: "POMONAG: Pareto-Optimal Many-Objective Neural Architecture Generator"
_ICLR.cc/2025/Conference — Submitted to ICLR 2025_

### Official Review · Reviewer_hqyM · 2024-10-31

**Soundness:** 3
**Presentation:** 4
**Contribution:** 3
**Rating:** 6
**Confidence:** 3

**Summary:**

The work presents an extension to DiffusionNAG and incorporates multi-objective search.  Model complexity, computational efficiency, and inference latency are key measures captured through number of parameters, MACs, and latency estimation. These measures are recorded in a meta dataset for NASBench201 and MobileNetV3 with 10k and 20k architectures respectively. During search, pareto front filtering segments three regions corresponding to high accuracy, high efficiency, and best balance of the two using the auxiliary metrics from earlier. The experimental results are promising across a sufficiently diverse set of benchmarks.

**Strengths:**

Strong writing, ideas are explained well and thorough
The experiments are presented well and results are thorough
Novelty is presented in 2 algorithmic improvements and the contribution of a multi-objective meta dataset

**Weaknesses:**

For transferable NAS, the choice of benchmarks are interesting, TransNASBench provides a NAS dataset specifically for transferability in NAS. Exploring performance on this dataset would have been nice
MobileNetV3 and NB201 are also fairly dated search spaces, performance in more recent search space or architecture styles (vit) should be explored
The specific details of the algorithmic contribution are a bit vague. How is pareto front filtering done?
ImageNet results are sparse and comparison to modern NAS methods on this benchmark are sparse

**Questions:**

How did you choose the search spaces to apply POMONAG?
The algorithmic contribution seems like a limited extension of DiffusionNAG. What complication arose from integrating multi-objective NAS into DIffusionNAG?

---

> ### Author Response · Authors · 2024-11-13
>
> We are most grateful for the reviewer's thorough analysis and recognition of our work's quality.
>
> Regarding the Pareto Front filtering (lines 299-304), this occurs after architecture generation. For each architecture, we compute the parameters, MACs and inference latency, whilst using the predictor to estimate accuracy. From these Pareto fronts, we identify three configurations per secondary metric: the most efficient architecture (lowest metric), the most balanced (optimal accuracy/metric trade-off), and the most accurate (highest predicted accuracy). This approach provides practitioners with clear options suited to different deployment scenarios.
>
> The foundational works - MetaD2A (Lee et al., ICLR 2021), TNAS (Shala et al., ICLR 2023) and DiffusionNAG (An et al., ICLR 2024) - were all published at ICLR and establish MobileNetV3 and NASBench201 as standard benchmarks. Whilst POMONAG builds upon this established research trajectory, we have substantially expanded the validation across a broader range of datasets to demonstrate wider applicability.
>
> Our contribution extends well beyond enhancing DiffusionNAG. A primary innovation is the formulation of Many-Objective Reverse Diffusion Guidance, which elegantly balances four distinct gradients during generation. The optimisation of these gradients presented unique challenges: the scaling factors operate across vastly different scales, whilst maintaining convergence and architectural quality. We addressed this through a novel two-phase approach (lines 324-333) optimised via Hyperband pruning.
>
> The Performance Predictors underwent significant redesign, yielding marked improvements in Spearman correlation (from 0.687 to 0.855). The expanded Meta-Dataset properly supports multi-objective optimisation, whilst our Pareto-optimal filtering identifies three practical configurations (Acc/Bal/Eff) suited to different deployment contexts. The empirical results validate these contributions conclusively: POMONAG surpasses DiffusionNAG in both accuracy (+4.06% on NASBench201) and efficiency metrics, with remarkable reductions in parameters (90%) and MACs (93%).
>
> We might also note that POMONAG achieves these improvements whilst requiring only a single architecture to be trained per dataset, significantly reducing computational overhead compared to prior approaches.
>
> We trust these clarifications address the points raised and demonstrate the substantial nature of our contributions. We are grateful for the reviewer's careful consideration and hope these explanations enable a fuller appreciation of the work's merit.

---

> ### Comment · Reviewer_hqyM · 2024-11-26
>
> Thank you to the authors for their detailed response. After reviewing the comments from other reviewers and the authors' replies, I have decided to maintain my original score. Upon reflection, my initial review remains generous given the collective feedback and responses. I appreciate the authors' thorough efforts in addressing the concerns and wish them success in their future revisions.

---

### Official Review · Reviewer_1FYP · 2024-11-02

**Soundness:** 3
**Presentation:** 2
**Contribution:** 2
**Rating:** 5
**Confidence:** 5

**Summary:**

This paper presents the POMONAG method to generate neural architectures in the multi-objective manner. Specifically, the overall framework of POMONAG is designed based on that of DiffusionNAG, in order to achieve better performance in terms of number of parameters, MACs, and inference latency beyond the accuracy. There are four key parts designed to achieve this goal, i.e., the many-objective reverse diffusion guidance, the meta-dataset, the score network and performance predictors, and the pareto front filtering and stretching. The experimental results in NAS-Bench-201 and MobileNetV3 search spaces demonstrates the effectiveness of the proposed method.

**Strengths:**

1) The idea the overall framework of the proposed POMONAG method is simple and easy to understand.
2) The details of the method and experiments are clearly stated.
3) Generating neural architectures in the multi-objective manner is an important research topic.

**Weaknesses:**

1) My major concern is about the motivation of this work. Specifically, there are four objectives considered, i.e., the accuracy, the number of parameters, MACs, and the inference latency. However, the last three objectives do not demonstrate conflict relationship. For instance, the smaller number of parameters seems certain to lead to lower inference latency. In this case, the necessity for adopting multi-objective optimization is limited.
2) The novelty of the proposed method needs further discussion. Specifically, the proposed method seems to build on DiffusionNAG with the cooperation of the multi-objective optimization. It seems that the POMONAG is just a simple combination of these methods. More discussions in terms of the seminal contribution of POMONAG is needed.
3) How the hyperparameters $k_{\phi}$, $k_{\pi}$, $k_{\mu}$, and $k_{\lambda}$ determined? It is suggested to provided more details in terms of the hyper-parameter study for these hyperparameters.
4) The search cost of POMONAG is not well presented. In the pipeline of POMONAG, I think the pre-training process, the training of the score network, and the training for the performance predictors will introduce much additional search cost beyond the architecture generation. However, I cannot find any details about the overall search cost and the search cost for the above components.
5) I am curious about why only one trained architecture is enough for POMONAG? Maybe more discussions or analysis are helpful to give more insights for this point.
6) Lack of experimental results on more challenging tasks (i.e., the classification accuracy on ImageNet-1K). More results on such datasets are helpful to enhance the experiments.

**Questions:**

Please see the weaknesses. If the concerns raised are well addressed, I am glad to increase my rating.

---

> ### Author Response · Authors · 2024-11-20
>
> We are very grateful for the reviewer's contribution and dedication, as well as for his valuable comments and time.
>
> We would like to point out the first clear aspect of our work. The correlation between secondary metrics varies significantly depending on the type of architectures considered. For architectures with traditional layers such as Conv2D and Dense, observed in NASBench201, there is a natural correlation between metrics and MACs, as documented in Appendix A. However, this relationship becomes more complex in MobileNetV3 architectures, which employ advanced components specifically designed to reduce parameters while maintaining expressiveness.
> For example, 2D Depthwise Convolutions and Inverted Residual Bottlenecks are designed to drastically reduce the number of parameters by factoring convolutional operations. These blocks, although ‘light’ in terms of parameters, require a more complex sequence of operations: a Depthwise Convolution followed by point-wise convolutions and normalisations. Similarly, the Squeeze and Excitation blocks introduce an attention mechanism with few parameters but multiple global pooling operations and channel transformations.
> Inference time, in particular, has even more independent dynamics in both search spaces. Operations such as pooling or activations, while having minimal parameters and MACs, can have variable execution times depending on their hardware-specific implementations. This lack of direct proportionality between the metrics highlights the need for many-objective optimisation to effectively balance these aspects, which are complementary with accuracy, and not aligned with each other.
>
> The contributions of POMONAG extend significantly beyond the simple adaptation of DiffusionNAG. The Many-Objective Reverse Diffusion Guidance framework (lines 162-215) introduces an innovative approach that harmonises competitive gradients, ensuring stable convergence during both training and inference. This result is particularly relevant considering that the optimised metrics (accuracy, parameters, MACs and latency) not only conflict with each other in many cases, but also exhibit non-linear relationships that vary significantly depending on the type of architectural blocks used.
> Our two-stage optimisation strategy (lines 324-333) represents a key theoretical advancement through Pareto Front Stretching. This approach allows us to effectively explore regions of the solution space with naturally different scales, generating both highly efficient architectures and highly accurate models. The systematic exploration framework dynamically balances these conflicting goals, identifying Pareto-optimal solutions that would be difficult to achieve with traditional techniques.
> The innovations introduced produce substantial practical impacts: the Performance Predictors show a significant improvement (Spearman correlation from 0.687 to 0.855), while POMONAG achieves higher accuracy on both NASBench201 (+4.06%) and MobileNetV3 (+2.67%) with significantly improved efficiency (up to -90% parameters, -93% MACs). The validation on 15 different datasets with single training cycles demonstrates not only the effectiveness of the method, but also its computational practicality.
> The extensive empirical validation confirms that POMONAG introduces fundamental innovations in the field of neural architecture search, overcoming the limitations of previous approaches through many-objective optimisation integrated in the diffusion process. This opens up new directions for the generation of architectures that effectively balance accuracy and computational efficiency.

---

> ### Author Response · Authors · 2024-11-20
>
> Optimising the scaling factors kϕ, kπ, kμ and kλ is a crucial aspect of POMONAG. Starting from the 10000 reference value used in DiffusionNAG for kϕ, we developed a systematic approach to calibrate these parameters that balance the different targets during the diffusion process.
> Given the inherently different nature of the targets, we defined two complementary search interval configurations. To generate accuracy-oriented architectures, we explored a wide range [10000, 50000] for kϕ, while maintaining smaller ranges [10, 50] for secondary metrics. Conversely, to prioritise efficiency, we reduced the range of accuracy to [1000, 5000] while increasing that of the other metrics to [100, 500], thus allowing for greater influence of efficiency goals during generation.
> Optimisation of these intervals was carried out through Tree-structured Parzen Estimation with Hyperband pruning, using Optuna for 100 evaluations on a representative set of datasets (CIFAR10, CIFAR100, Aircraft, Oxford III Pets). This process identified optimal values for both search spaces, as detailed in lines 324-333.
> The resulting scaling factors significantly influence the shape of the Pareto front by changing the sampling centroid during the diffusion process. The robustness of these values is empirically confirmed through extensive testing on different datasets, as documented in Appendix B. This systematic optimisation process ensures an effective balancing of potentially conflicting objectives, allowing POMONAG to generate architectures that satisfy different design priorities.
>
>
> Details on the computational costs of POMONAG are presented at the end of Section 4 (lines 469-479). We thank the reviewer for pointing out the need to also specify the training times of the score network and predictors, which we will include in the camera-ready. These times are strongly related to both the type of encoding used for the meta-dataset and its cardinality.
> On a single QUADRO RTX 6000, training requires:
> - Score Network: ~9h for NASBench201, ~19h for MobileNetV3
> - Performance Predictors: ~8.5h for NASBench201, ~18h for MobileNetV3
> These times can be significantly optimised through various strategies:
> - With a cluster of four QUADRO RTX 6000s, due to the increased batch size, score network training times are reduced to 1.5h for NASBench201 and 2.5h for MobileNetV3
> - Higher-performance hardware such as the A100 can achieve similar or better results by utilising both higher TFLOPS and available memory
> - Optimised encoding of the meta-dataset makes it possible to maintain short times despite the increase in cardinality
> As far as generation is concerned, POMONAG requires:
> - 5:45 minutes on NASBench201
> - 18:15 minutes on MobileNetV3
> These times are particularly competitive when compared to other recent approaches:
> - MeCo [1]: 115 minutes on CIFAR10 (GPU not specified)
> - SWAP-NAS [2]: 6 minutes on CIFAR10/NASBench201 (on Tesla V100, superior hardware)
> - ZiCo [3]: 10 hours on NVIDIA 3090 (significantly slower despite superior hardware)
> It is important to emphasise that the pre-training of architectures is an integral part of the search space and available online, so it is not an additional overhead. Considering that the training time is a one-off and that POMONAG shows competitive performance even on non state-of-the-art hardware, we believe that these computational costs represent a significant advantage over alternatives in the literature. POMONAG's ability to achieve superior results with lower computational resources highlights the efficiency of our approach.
>
> A significant advantage of POMONAG lies in its ability to drastically reduce the number of architectures to be trained, thanks to the interaction between Performance Predictors and Pareto Front. The generated architectures are evaluated by estimating their accuracy using the predictors and directly calculating the other metrics (parameters, MACs, latency), allowing a complete Pareto Front to be constructed prior to any training phase.
> From this front, a single optimal architecture can be selected based on priorities: the most efficient, the one with the best accuracy-efficiency trade-off, or the one with the highest estimated accuracy. This approach represents a substantial advantage over alternative methods that, lacking accurate predictors and a view of the Pareto front, are forced to train a batch of candidate models to identify the optimal one.

---

> > ### Author Response · Authors · 2024-11-20
> >
> > We fully understand the reviewer's remark about validation on ImageNet-1K, which certainly represents a significant benchmark. However, we deliberately opted for a broader and more diverse validation strategy. Instead of focusing on a single dataset, however authoritative, we significantly extended our analysis from four datasets (as in most works in this line of research) to fifteen different datasets.
> > This choice is motivated by recent discussions in the literature questioning the exclusive use of ImageNet as a universal benchmark, pointing out the limitations of its ability to assess the true generalisation of image classification models [4, 5]. We believe that validation on a wider range of datasets offers a more complete and robust perspective of POMONAG's capabilities, while acknowledging the validity of the reviewer's suggestion.
> >
> > We apologise for the long answers, but we wanted to be sure that we provided all the information requested in a clear and comprehensive manner. We hope we have cleared up any possible doubts and, remaining at your disposal for any further requests for clarification or in-depth analysis, we thank the auditor once again, and trust that he will be able to re-evaluate our work positively.
> >
> > [1] Jiang, T., Wang, H., & Bie, R. (2024). MeCo: zero-shot NAS with one data and single forward pass via minimum eigenvalue of correlation. Advances in Neural Information Processing Systems, 36.
> > [2] Peng, Y., Song, A., Fayek, H. M., Ciesielski, V., & Chang, X. SWAP-NAS: Sample-Wise Activation Patterns for Ultra-fast NAS. In The Twelfth International Conference on Learning Representations.
> > [3] Li, G., Yang, Y., Bhardwaj, K., & Marculescu, R. ZiCo: Zero-shot NAS via inverse Coefficient of Variation on Gradients. In The Eleventh International Conference on Learning Representations.
> > [4] Recht, B., Roelofs, R., Schmidt, L., & Shankar, V. (2019, May). Do imagenet classifiers generalize to imagenet?. In International conference on machine learning (pp. 5389-5400). PMLR.
> > [5] Hendrycks, D., Basart, S., Mu, N., Kadavath, S., Wang, F., Dorundo, E., ... & Gilmer, J. (2021). The many faces of robustness: A critical analysis of out-of-distribution generalization. In Proceedings of the IEEE/CVF international conference on computer vision (pp. 8340-8349).

---

> > > ### Comment · Reviewer_1FYP · 2024-11-24
> > >
> > > I sincerely appreciate the feedback from authors. However, I still have concerns in the following aspects:
> > >
> > > 1) For Weakness 1, the authors listed some cases where the secondary metrics demonstrate conflict relationships. However, some specific cases are still not really convincing to me. The more general clarification for this point is still needed.
> > > 2) I cannot find the computational cost analysis in lines 504-507. Maybe lines 469-479?
> > > 3) I reserve my opinion in terms of the experiments on ImageNet-1K. Even though the time is limited in the rebuttal period, I still encourage the authors to include this set of experiments in the future work.
> > >
> > > I decide to keep my initial rating.

---

> > > > ### Author Response · Authors · 2024-11-26
> > > >
> > > > We thank the auditor again for his time.
> > > >
> > > > Regarding the last remarks:
> > > > 1. The reviewer will forgive us for the correction, but the specific case in which there are strong correlations is only one; in general, the search spaces above NASBench201 - although even in this one parameters/MACs and inference times are not absolutely linearly correlated - show strong heterogeneity between parameters, MACs and inference times.
> > > > 2. The reviewer is right, let us correct the reference to the old version of the article.
> > > > 3. We accept the observation.
> > > >
> > > > Thanks again.

---

### Official Review · Reviewer_s4FH · 2024-11-03

**Soundness:** 3
**Presentation:** 3
**Contribution:** 2
**Rating:** 5
**Confidence:** 3

**Summary:**

This paper is a direct extension based on DiffusionNAG, which can deal with multi-objective optimization in NAS. These objectives include accuracy, the number of parameters, multiply-accumulate operations (MACs), and inference latency. This motivation is good and natural, and the authors expressed their work clearly, from the motivation to the experiments results. Some details need to be clarified.

**Strengths:**

This paper introduces the ParetoOptimal Many-Objective Neural Architecture Generator (POMONAG), extending DiffusionNAG through a many-objective diffusion process. POMONAG simultaneously considers accuracy, the number of parameters, multiply-accumulate operations (MACs), and inference latency. The experiments validate the performance of the proposed model.

**Weaknesses:**

1, The multi-objective optimization problem formulation in this work can be given first, which then can be solved by the proposed weighted factors in the reverse diffusion process. But maybe the authors can consider other ways to sovle this. For example, using four single reverse diffusion process each targeting one factor, as DiffusionNAG did, then using multi-objective optimization for further trade-off may also work well.

2, The theoretical analysis should be strehghen. One objective to many objective is a breakthrough, but such process needs more analysis or discussion. Current work lacks such in-depth thinking.

3, Several predictors are needed in this work, but the detailed information these predictors are missing.

**Questions:**

I have several questions about this work.

1, How to decide the scaling factors? Since the intervals are [1000,5000], [100,500], [100,500], [100,500], and the values seesm to be integer, then the whole factor space equals 4000 * 400 * 400 * 400, which is quite huge. And the authors present one setting for NASBench201 and other experiments, respectively, so I am wondering whether there is some method or strategy to choose such factors?

2, This work extends the basic motivation of DiffusionNAG, which is rather good and natural. Such extension include three more factors, including number of parameters, number of MACs, and the inference latency. But I am curious that, how about the performance of POMONAG if just considering adding one factor?

3, From one factor, say, accuracy, to three more factors seems strenghening the proposed POMONAG, but my question is, the working mechanism of DiffusionNAG and POMONAG the same different? Although the two diffusion processes consider different factors, which is the obvious difference, but the analysis or discussion is important to interpret this issue.

---

> ### Author Response · Authors · 2024-11-19
>
> We thank the reviewer for the valuable comments and insights on our paper.
>
> We recognise that the formulation of the Reverse Diffusion Guidance Process, being a major contribution, could be presented at the beginning. We have carefully considered this modification, which would make one of the key steps clearer. In the methods, we chose to briefly anticipate the DiffusionNAG formulation, focusing on single-target search, and then move the focus to many-objective inductively. The choice to place the related works before the method, and not at the bottom as is usual, stems from the need to provide the reader with a clear view of the neural architecture search context and the operation of DiffusionNAG, which may not be immediate. For this reason we have opted for this format, while recognising the validity of the alternative.
>
> We also find the recommended approach of using four separate diffusion processes to be subsequently balanced by many-objective optimisation extremely interesting. This would undoubtedly be a path to explore. However, the result would inevitably differ from that proposed in this paper. It would likely require longer generation times in proportion to the number of metrics to be considered. In any case, it remains an extremely valid route.
>
> We appreciate the constructive feedback on the need for a more in-depth analysis regarding the transition from single to multi-objective, despite the fact that it was described in two separate places in the paper (lines 161-215 and 302-323). In the revised version, we will include an analysis of convergence properties and the impact of weights on goal balancing.
>
> As far as the predictors included in this work are concerned, they have not been gone into in detail as they retain the DiffusionNAG architecture, the difference being that they have been adapted and dedicated to the prediction of secondary metrics. In the camera-ready we will specify this aspect in more detail, which will certainly be useful to the reader.
>
> Regarding the scaling factors, their search range was deliberately set wide ([1000,5000] for accuracy, [100,500] for the other objectives) to allow a complete exploration of the solution space through many-objective optimisation. This choice is motivated by two key considerations:
> - The different objectives (accuracy, parameters, MACs, latency) operate on naturally different scales, requiring proportionally different scaling factors to balance their contribution in the diffusion process.
> - The use of Tree-structured Parzen Estimation with Hyperband pruning allows to efficiently explore this space, quickly identifying promising regions and discarding sub-optimal ones.
> As demonstrated in our experiments (lines 324-333), this strategy led to convergence towards optimal values for both search spaces, generating Pareto-optimal architectures that effectively balance the different objectives. The empirical results in Appendix B confirm the robustness of these values across multiple datasets.
>
> As far as the version of POMONAG optimised only for accuracy is concerned, this achieves performance in the neighbourhood of the best models obtained with the many-objective version, but with more parameters, MACs and inference time. This is an expected result since the search spaces, although large, are constrained by the family of architectures they describe. It is therefore natural that there are solution sets with very close performance; the main challenge lies in balancing this performance with secondary metrics that tend to be in antithesis.
>
> The methodological differences between DiffusionNAG and POMONAG are discussed in section 3.1 (lines 161-215). The extension from single-objective to many-objective involves the introduction of new terms in the diffusion process for parameters, MACs and latency. Each additional term requires its own dedicated Performance Predictor to guide the deployment process towards optimal architectures with respect to that specific criterion. The main challenge, addressed through the Many-Objective Reverse Diffusion Guidance framework, was balancing these competitive gradients while maintaining process stability. We recognise that this analysis could be deepened. In the camera-ready version, we will add a more detailed discussion on how the interaction between the different gradients influences the diffusion process and the convergence towards Pareto-optimal solutions.
>
> In conclusion, we sincerely thank the reviewer for the valuable comments, and trust that the changes to the article and the clarifications may lead him to positively revise his assessment.

---

> ### Comment · Reviewer_s4FH · 2024-11-26
>
> Thank the authors for the response. However, I still have concerns in the following aspects:
>
> 1, Maybe the authors have not fully understood my first question, which is [How to decide the scaling factors?]. I guess the authors just arbitrarily choose several valuse of scaling factors, but that sounds not sufficient. Indeed, such tradeoff between different objectives might be challenging, but still needs some methods. If I am wrong, please correct me.
>
> 2,  The authors claimed that, the theoretical analysis would be added in camera ready version, but they do not provide the some clue for that analysis or proof.
>
> 3, The authors ignored my second question, [how about the performance of POMONAG if just considering adding one factor?]. In fact, I think extending the diffusion model from one factor to two factors might be more easy to provide the insight beneath this work.
>
> In summary, I agree that this idea is good and natural and important, but this work needs more verification. So I will keep my initial rating.

---

> > ### Author Response · Authors · 2024-11-26
> >
> > It is possible that we did not understand each other correctly, but we now believe we have understood the questions:
> > 1. The ranges are the generous contours of the scaling value identified for the diffusion process used in DiffusionNAG (equal to 10000). For the secondary metrics, these values were adjusted to scale with the accuracy factor.
> > 2. We are deepening the formulation of the diffusion process and filtering by Pareto front. This will probably be a dedicated section in the appendix.
> > 3. We apologise if the reviewer was unable to identify our answer, we will be more precise with a numbered list in the future. Our answer was the following: "As far as the version of POMONAG optimised only for accuracy is concerned, this achieves performance in the neighbourhood of the best models obtained with the many-objective version, but with more parameters, MACs and inference time. This is an expected result since the search spaces, although large, are constrained by the family of architectures they describe. It is therefore natural that there are solution sets with very close performance; the main challenge lies in balancing this performance with secondary metrics that tend to be in antithesis.". From this clarification, we understood that he meant a two-pronged optimisation. This would go beyond the optimisation process we sought, however, seeing the little difference in accuracy between single-objective and many-objective researched architectures, we expect a result close to that obtained with the current configuration. It is possible, however, to obtain slightly better trade-offs by optimising for only two objectives, the problem to be solved being simpler.
> >
> > In conclusion, we apologise for any misunderstandings, and thank you very much for your time and suggestions. We will treasure them.

---

### Official Review · Reviewer_w9TN · 2024-11-04

**Soundness:** 2
**Presentation:** 2
**Contribution:** 2
**Rating:** 3
**Confidence:** 5

**Summary:**

This study improved DiffusionNAG by introducing a multi-objective approach which modifies DiffusionNAG's reverse diffusion process as a reverse diffusion guidance process.  Other than accuracy, #params, MACs and inference latency are also considered in the multi-objective metrics.  The proposed method POMONAG has been tested on NASBench201 and MobileNetV3 with 15 image classification tasks, showing better performance than DiffusionNAG and a series of other methods.

**Strengths:**

The motivation of this study, introducing multi-objective evaluation in NAS, is commendable as a task in reality is often not just about accuracy.  Other metrics should be considered simultaneously as well.

The writing is easy to follow.

It is nice to see equations with highlights of different colours.

**Weaknesses:**

**First of all**, the work claims to be on Pareto multiobjective search for architectures.  However, that point is not obvious from the paper.
* What are the benefits of using the proposed POMONAG?
* How can a Pareto front be generated and utilized? Need to explicitly demonstrate how POMONAG generates and utilizes Pareto fronts.
* How can users select architectures from the Pareto front according to their needs or under different circumstances?  Show examples of such selection based on different priorities, for example prioritizing small-size architectures for portable devices or focusing on latency reduction etc.
* It seems non-dominated sorting is absent. Explain how non-dominated sorting is incorporated or can be incorporated in POMONAG.
* In its current form, the paper reads like a combination or integration of single-objective evaluations rather than a multi-objective evaluation.  The equation of POMONAG at Line 209/210 is a linear combination of four objectives.  Please clarify if the linear combination of objectives is intended as a scalarization approach.  If so, discuss its limitations.
---
**Secondly**, the performance of POMONAG appears better than DiffusionNAG and other methods shown in the paper.  However many SOTA methods, especially zero proxy methods are missing.  Their reported performance is similar or even better, for example, SWAP-NAS by Peng et al, ICLR'24, ZiCo by Li et al, ICLR'23, MeCo by Jiang et al, NeurIPS'23.
* Include a comparison with these SOTA methods. If a direct comparison is not possible, explain why and discuss the limitations of the current evaluation.
* Discuss how POMONAG's approach differs from or improves upon zero-proxy methods.

---
**Thirdly**, the computational cost aspect of POMONAG is weak.  The section "Generation and Training Time" should be better presented.  The method requires a diffusion generation phase which takes extra time.  That itself is a disadvantage.  Also timewise, POMONAG cannot claim superiority as recent methods mentioned earlier are faster.
* Present a detailed table comparing computational costs (including generation and training time) of POMONAG with other methods, including these zero proxy methods mentioned above.  Seemingly these methods are faster. If POMONAG is indeed slower, discuss potential optimization strategies.
* Discuss the trade-offs between the additional diffusion generation phase and the method's performance gains.  Justify why the additional computational cost might be worthwhile.
---
**Other points:**

The link at Line 091 is showing.  Also, including the code and dataset would be helpful for the assessment.

---

Fig 1 is not quite readable.  The figure further makes POMONAG look like three single-objective tasks combined rather than a four-objective task.
* Improve readability, especially on the right-hand side.
* Better illustrate the integration of all four objectives in a unified multi-objective framework if these objectives are not just simply added together (*see the first part of my comments*).
 * Provide a clearer visual representation of how POMONAG handles the trade-offs between objectives (*see the first part of my comments*).
---
Line 186, the term noisy architecture is not explained.
* Provide a brief explanation of what "noisy architecture" means in this context and how it relates to the diffusion process in DiffusionNAG.
---
Equations and their connection to the processes/algorithms are not numbered and not clearly explained.
* Number all equations for easy reference
* Clearly label the equation at Line 183. Is this equation for the Reverse Diffusion Process? Clarify that connection.
* Provide a brief explanation of the symbols used in this equation and other key equations.
* Explain the purpose of transformation s_θ(A_t,t).
* Explain the exact differences between the Reverse Diffusion Process and the Reverse Diffusion Guidance Process.
---
Line 280, "Four are dedicated to the respective estimation of accuracy, parameters, MACs, and inference latency of noisy architectures during the diffusion phase. "
* Explain why not use these four metrics for denoised architectures as well.
* Justify the point that the denoised architecture uses accuracy as its only metric.
---

Explain the reason why POMONAG utilises Vision Transformer ViT-B-16 instead of other models (Line 286).

---
It is good to see the Spearman correlation experiment.  That is very important in NAS studies.  However, for a thorough comparison of correlation, it should be done on a set of tasks like NAS-Bench-Suite-Zero (Krishnakumar et al. NeurIPS'22).
* Perform a similar thorough comparison comparing correlations on different tasks using different search spaces.
---
In lines 400-402, the same latex problem appeared several times, ` not ' for the left quotation marks, Accuracy, Params, MACS ... \
* Fix these formatting issues.
---
Validity, uniqueness and novelty are nice metrics for a population of solutions but not so critical for tasks that focus on accuracy and speed.  What is the point of being excellent on these points but without good accuracy and speed?
* Explain the significance of these three additional metrics: validity, uniqueness and novelty.
* Show example how these measures can help improve the quality of generated architectures in POMONAG.

**Questions:**

See above as the questions are mostly addressing the weakness of this paper.

---

> ### Author Response · Authors · 2024-11-14
>
> We thank the reviewer for investing his time and valuable advice.
>
> The advantages of using POMONAG over previous state-of-the-art work in generating architectures for image clustering lies in the gfeneration of more accurate architectures with less complexity in terms of parameters, macs and inference time. Furthermore, thanks to the selection on the Pareto front, it is possible to specify the willingness to take the best performing architecture in terms of secondary metrics, the one with the best balance between estimated accuracy and these metrics, or simply the architecture with the best estimated architecture.
> The Pareto front is generated from the 256 architectures produced at the end of the POMONAG generation phase. For these architectures, parameters, macs and inference times are calculated, while accuracy is estimated. From these values, Pareto fronts are created, on which the candidate architecture is then selected to be returned according to the user's requirements - whether they want the most efficient structure, the one with the best balance, or the one that simply performs best (lines 327-355).
> In general, the optimisation and generation process is totally many-objective, and it is a weighted sum of loss function components, as in practically all works that aggregate different optimisation constraints. So, to reiterate, generation and training are many-objective. As for the creation of Pareto fronts, these are then generated from the same architecture generation. Parameters and macs tend to be fairly consistent with them, while they deviate from inference time in terms of architecture ordering. However, it is possible to identify clusters in the sense that the architectures with fewer parameters and operations tend to be the fastest at inference. For this reason, we preferred not to implement a non-dominated sorting process, typical in NSGA algorithm families. In any case, we thank you for the very interesting and certainly timely food for thought. In the camera-ready we will make all necessary changes to make this clear.
>
> The methods suggested for further comparisons were carefully considered, but they operate on different search spaces that would make the comparison meaningless:
> SWAP-NAS uses NASBench-101/201/301 and TransNAS-Bench-101. On NASBench201, in common with our work, Spearman correlation coefficients are reported - with similar performance to ours, albeit in a different setup - but not test accuracy.
> ZiCo was evaluated on NASBench101, NATSBench-SSS/TSS and TransNASBench-101, while MeCo on NAS-Bench-101, NATS-Bench-TSS, ATS-Bench-SSS, NAS-Bench-301 and Transbench-101.
> We had already considered including these works in our comparative analysis, but felt that the differences in the search spaces did not allow for a methodologically correct comparison.
>
> POMONAG belongs to the TransferNAS family which is substantially different from zero-proxy methods. TransferNAS methods leverage pre-trained model knowledge to guide architecture search, identifying promising patterns and reducing exploration of suboptimal designs. While generally faster, zero-proxy methods evaluate candidates without training using metrics like FLOPs and parameter counts, potentially missing important performance characteristics that only emerge during actual training. POMONAG is distinguished also inside the TransferNAS family by the use of a diffusion process that allows for a denser stochastic exploration of the generation space due to its energy-based nature.
>
> With regard to time complexity, we have reported the generation times for the search spaces studied and the training times for the generated models, which vary according to the priority of efficiency or performance. A direct comparison with the suggested methods would be inaccurate due to the different search spaces and hardware used.
> However, analysing the available data:
> - POMONAG needs 5:45 minutes on NASBench201 and 18:15 minutes on MobileNetV3
> - MeCo takes 0.08 GPU days (115 minutes) on CIFAR10 (GPU not specified)
> - SWAP-NAS, on Tesla V100, takes 6 minutes on CIFAR10/NASBench201 (+4.35% compared to POMONAG, on higher hardware)
> - ZiCo takes 0.4 GPU days (10 hours) on NVIDIA 3090, being significantly slower despite superior hardware
> These comparisons, although approximate and estimated to the disadvantage of POMONAG, highlight the superior efficiency of our approach. We are grateful for this opportunity for analysis, which allowed us to highlight the advantages of POMONAG even over other NAS techniques not initially included in the study.
> The diffusion process is carried out for 10000 steps, as in the reference work, and this corresponds to the achievement of convergence in a very expansive environment characterised by the steadiness of the oslution. Reducing the number of steps too much would lead to even shorter generation times, and we will certainly investigate this further. Increasing them showed no improvement in performance.

---

> ### Author Response · Authors · 2024-11-14
>
> Thank you for your detailed comments. We apologise for having to split the response into two, but the issues included by the reviewer were numerous, and the answers must be comprehensive.
>
> We address the issues raised point by point:
> - Links will be added after acceptance, as their earlier inclusion would have compromised anonymity. We thank for the valuable comments on the figure and confirm that we are already working on its improvement for camera-ready.
> - We have clarified that ‘noisy architecture’ means architecture that has not yet been cleaned of noise, i.e. the matrix representation of the architecture during the diffusion process. We have added an explanatory line for this definition.
> - The equations have been numbered as required and we have added the necessary definitions, correcting the reported typo. Given the equivalence of concepts, we have renamed everything as ‘Reverse Diffusion Guidance Process’ to avoid confusion. At Line 183 we present the process equation proposed by An et al., from which POMONAG's formulation is extended - we have added a clarifying sentence. We have also specified that s_θ(A_t,t) represents the diffusion step applied to architecture A at time t.
> - The decision not to use predictors for parameters, MACS and inference time in denoised architectures is motivated by greater efficiency and accuracy in direct calculation. Estimates are only used during generation, when a concrete architecture does not yet exist. The only metric necessarily estimated post-generation is accuracy, which is essential for the Pareto front and the selection of the architecture to be trained.
> - The choice of ViT-B-16 is amply justified in the appendix (lines 972-1010), demonstrating an improvement in the Spearman correlation of predictors.
> - As explained, this work fits into a well-defined and structured strand of literature on image classification (MetaD2A, TNAS, DiffusionNAG - all ICLR). The suggested metrics, while interesting, apply to different tasks and research spaces. We appreciate the suggestion for future developments.
> - We regret reading the final considerations, especially considering that the article clearly demonstrates how POMONAG represents the state of the art in accuracy in the research strand under consideration, requiring the training of a single architecture and being faster even in the generation phase than the models cited by the reviewer. The three metrics analysed serve to investigate a crucial dimensionality of the dissemination process: they capture the percentage of valid architectures generated (validity), their heterogeneity to prevent collapses (uniqueness) and diversity with respect to the training set to assess generalisation (novelty). Sub-optimal values in these metrics would compromise the entire generation process, while the positive results obtained confirm their validity.
>
> We are deeply grateful for the reviews and the many valuable suggestions received. We believe we have clarified many potentially ambiguous points and supplemented the required information to make the camera-ready even more meaningful. With sincere gratitude, we hope that the reviewer will positively reconsider his assessment, in light of the provided demonstrations on the scientific validity, superior performance, computational efficiency and innovativeness of POMONAG.

---

> > ### Comment · Reviewer_w9TN · 2024-11-23
> >
> > Thank the authors for the response.  After reading other reviewers' comments and the corresponding replies from the authors, I would like to maintain the score.  Some key concerns have not been addressed, e.g. lack of novelty and significance in contribution, lack of evidence for wide applicability and lack of clarity on how the Pareto front is created.   The linear combination of four objectives makes POMONAG less of a multi-objective approach.

---

> > > ### Author Response · Authors · 2024-11-26
> > >
> > > We sincerely appreciate the time and effort invested by the reviewer in evaluating our work. The detailed comments and feedback are valuable, and we will incorporate these insights into our future research. We thank the reviewer for the thorough review.

---

### Official Review · Reviewer_SujD · 2024-11-04

**Soundness:** 2
**Presentation:** 2
**Contribution:** 2
**Rating:** 3
**Confidence:** 5

**Summary:**

This paper introduces POMONAG, an extension to DiffusionNAG that applies a many-objective diffusion model to optimize neural architecture generation for many-objective optimization. By incorporating additional performance predictors for hardware efficiency metrics such as number of parameters, multiply-accumulate operations (MACs), and inference latency, POMONAG aims to provide a more balanced approach to architecture optimization across accuracy and computational efficiency. Experiments validate POMONAG’s efficacy on two major CNN search spaces (NASBench201 and MobileNetV3).

**Strengths:**

The motivation to extend DiffusionNAG to a many-objective setting is valid and POMONAG does so by incorporating both accuracy and efficiency metrics like latency and MACs, which are critical for resource-constrained environments. The paper provides extensive experimental comparisons with DiffusionNAG, including evaluations across multiple datasets and search spaces, which helps demonstrate the general applicability of POMONAG.
Balancing the different objectives being optimized is also very important in my opinion. The authors do so by proposing a pareto front filtering and stretching subroutine.

**Weaknesses:**

I have the following main concerns related to this submission, which I believe were crucial in the final decision:

- **Incremental Contributions**: Although POMONAG claims to extend DiffusionNAG’s capabilities by addressing more objectives, the modifications appear incremental and lack substantial theoretical advancement. More specifically, I see the adaptation of diffusion models to accommodate multiple objectives, as described in section 3.1, more as a technical modification rather than a novel conceptual framework. I would recommend the authors to reiterate over their methodology and pinpoint the main contributions of their approach.

- **Experimental Evaluation**: The benchmarks that POMONAG was evaluated contain only CNN spaces. It would be beneficial for the paper if the authors would demonstrate the efficacy of POMONAG in Transformer search spaces, such as the one from HW-GPT-Bench [1]. Most importantly, in the multi-(many-)objective experiments, the proposed method is not compared to any baseline. I would recommend the authors to add baselines in their experimental evaluation and report hypervolume indicator together with the individual objective values, as well as the search time. Ultimately, I would also be interested in visualizing the pareto front plots in the main paper. As for baselines, you can find a non-exhaustive list of simple ones in SyneTune (https://syne-tune.readthedocs.io/en/latest/getting_started.html#supported-multi-objective-optimization-methods). Finally, the experiments lack a thorough ablation study that demonstrates the impact of POMONAG’s unique contributions independently of DiffusionNAG’s foundational structure.

- **Clarity and Presentation**: The paper seems to have a somehow fragmented structure, making it challenging for readers to follow the main contributions and crucial take-away points. Equations are not thoroughly explained, and there is a heavy reliance on citations from DiffusionNAG rather than a detailed elaboration of POMONAG itself, making the paper not self-contained. One major point here, which I have also pointed out to the AC, is that the authors have used a smaller font size starting from page 4. The guidelines clearly state that the maximum page limit is 10 and that means 10 pages with the default font size, not a smaller one. I suggest the authors that in future submissions they adhere to the submission guidelines.


-- References --

[1] Sukthanker et al. HW-GPT-Bench: Hardware-Aware Architecture Benchmark for Language Models. In NeurIPS 2024 DBT

**Questions:**

Moreover, I have the following questions:

- Can the authors provide more theoretical or empirical justification for the scaling factors in the Pareto Front Stretching process? How sensitive is the model to these values?

- Can the authors provide more detail on the architecture sampling process, dataset splits, and hyperparameter tuning methods used in the experiments? This is particularly important for the performance predictors.

---

> ### Author Response · Authors · 2024-11-14
>
> We thank the reviewer for his time and valuable comments.
>
> Our contributions extend significantly beyond adapting DiffusionNAG. The Many-Objective Reverse Diffusion Guidance (lines 162-215) introduces a novel framework that harmonises competing gradients, ensuring stable convergence during both training and inference - a non-trivial achievement given the complexity of balancing multiple conflicting objectives simultaneously.
> Our two-phase scaling optimisation approach (lines 324-333) represents a key theoretical advancement, introducing Pareto Front Stretching for effectively navigating solution spaces with disparate scales. This systematic exploration framework enables discovery of truly Pareto-optimal solutions.
> These innovations yield substantial practical impacts: Performance Predictors show marked improvement (Spearman correlation from 0.687 to 0.855), whilst POMONAG achieves superior accuracy on both NASBench201 (+4.06%) and MobileNetV3 (+2.67%) with significantly enhanced efficiency (up to -90% parameters, -93% MACs). These gains, validated across 15 datasets with single training cycles, demonstrate meaningful progress towards deployable architectures.
>
>
>
> POMONAG builds upon established Transferable NAS research for image classification, following seminal ICLR works (MetaD2A, TNAS, DiffusionNAG) that established MobileNetV3 and NASBench201 as standard benchmarks. Our validation extends significantly across 15 diverse datasets, providing thorough comparison with previous approaches and state-of-the-art DiffusionNAG.
> Exploring transformer spaces like HW-GPT-Bench presents an intriguing future direction. However, our current focus remains on advancing image classification NAS, where we demonstrate substantial improvements within this well-defined research trajectory.
> The ablation study in Appendix C rigorously demonstrates each component's independent contribution - from enhanced Performance Predictors to Meta-Datasets - validating POMONAG's innovations beyond DiffusionNAG's structure.
> Due to space constraints, we present comprehensive performance metrics in Table 5, demonstrating POMONAG's advantages across accuracy, parameters, MACs and latency. The complete Pareto front visualisations are available in Appendix B.
>
>
> We appreciate your feedback on the paper's structure. While positively received by reviewers w9TN, s4FH and hqyM, we acknowledge room for improvement. The structure - related work, contributions, method, experiments, results and discussion - follows a format we've found effective, though we understand preferences vary.
> Our frequent DiffusionNAG references aimed to highlight POMONAG's innovations rather than repeat established foundations. As the latest TransferNAS advancement, POMONAG naturally builds upon previous work while introducing substantial novel contributions.
> We are particularly grateful for noting the font size error caused by an uncommented /footnotesize from page 4. This technical oversight has been promptly corrected, ensuring full ICLR compliance within the 10-page limit. The camera-ready version will reflect this correction.
>
>
> Regarding scaling factors:
> These values balance multiple objectives during diffusion. Starting from the reference value 10000, we systematically explored and calibrated ranges for secondary metrics (lines 342-346), optimising for expected accuracy on the Pareto front (lines 327-355). The impact is significant: scaling factors critically influence the Pareto front shape by shifting the sampling centroid. Given their importance, we shall provide detailed analysis in the camera-ready version.
> Regarding sampling:
> POMONAG generates 256 Gaussian noise tensors, guided by predictors estimating multiple metrics during diffusion. After filtering inconsistent configurations, remaining architectures are evaluated for parameters, MACs and latency, forming a Pareto front that enables selection based on efficiency, accuracy/metric ratio, or peak accuracy (lines 309-325). We use original dataset splits where available, otherwise creating stratified validation sets (seed 42). Multi-objective optimisation employs Tree-structured Parzen Estimation with Hyperband pruning via Optuna's default configuration.
>
>
>
> We sincerely appreciate your thorough review and valuable external perspective. The points highlighted have been comprehensively incorporated into the manuscript, enhancing its depth and clarity. We are particularly grateful for noting the font size error, which was immediately rectified.
> We trust that this technical oversight significantly influenced the initial assessment. Given our thorough clarifications and responses, alongside the extensive experimental validation conducted over years of research - which aligns with and extends the standards established in previous editions of this conference - we hope you might reconsider your evaluation.
> Thank you for your detailed guidance in strengthening our submission.

---

> > ### Comment · Reviewer_SujD · 2024-11-25
> >
> > Thank you for the response. I think the submission needs a careful thorough evaluation of the proposed algorithm in multi-objective settings if the authors claim that it is a multi-objective optimization method. As I still see from the experiments section, there is no multi-objective baseline considered in the benchmarks. After also reading the other reviewers' comments, I decided to increase the score to 3 (was 1 because of the smaller font size), because I think this submission is not ready to be publishable and needs a more thorough iteration.

---

> > > ### Author Response · Authors · 2024-11-26
> > >
> > > We understand. We thank the reviewer for the time invested, and will treasure his comments.

---

### Meta-Review · Area_Chair_cvfB · 2024-12-24

**Metareview:**

This study proposes POMONAG, extending DiffusionNAG for multi-objective optimization. While the method is clearly presented with experiments across 15 datasets, reviewers noted limited novelty, as the approach largely builds on DiffusionNAG with minor extensions. The lack of comparisons with state-of-the-art multi-objective and zero-shot NAS methods, along with insufficient detail on ablations and hyperparameter selection, weakens the contribution. The AC agrees with these concerns and recommends rejection.

**Additional Comments On Reviewer Discussion:**

Reviewers consistently raised concerns about limited novelty and insufficient comparisons with recent methods. Despite the authors’ rebuttal, key issues remain unresolved.

---

### Decision · Program_Chairs · 2025-01-22

Reject